

# Bayesian RG flow in neural network field theories

Jessica N. Howard[1*], Marc S. Klinger[2†], Anindita Maiti[3‡] and Alexander G. Stapleton[4○]

**1** Kavli Institute for Theoretical Physics, Santa Barbara, CA USA
**2** Department of Physics, University of Illinois, Urbana, IL USA
**3** Perimeter Institute for Theoretical Physics, Waterloo, Ontario, Canada
**4** Centre for Theoretical Physics, Queen Mary University of London, London UK

* jnhoward@kitp.ucsb.edu , † marck3@illinois.edu ,
‡ amaiti@perimeterinstitute.ca , ○ a.g.stapleton@qmul.ac.uk

## Abstract

The Neural Network Field Theory correspondence (NNFT) is a mapping from neural network (NN) architectures into the space of statistical field theories (SFTs). The Bayesian renormalization group (BRG) is an information-theoretic coarse graining scheme that generalizes the principles of the exact renormalization group (ERG) to arbitrarily parameterized probability distributions, including those of NNs. In BRG, coarse graining is performed in parameter space with respect to an information-theoretic distinguishability scale set by the Fisher information metric. In this paper, we unify NNFT and BRG to form a powerful new framework for exploring the space of NNs and SFTs, which we coin BRG-NNFT. With BRG-NNFT, NN training dynamics can be interpreted as inducing a flow in the space of SFTs from the information-theoretic 'IR' → 'UV'. Conversely, applying an information-shell coarse graining to the trained network's parameters induces a flow in the space of SFTs from the information-theoretic 'UV' → 'IR'. When the information-theoretic cutoff scale coincides with a standard momentum scale, BRG is equivalent to ERG. We demonstrate the BRG-NNFT correspondence on two analytically tractable examples. First, we construct BRG flows for trained, infinite-width NNs, of arbitrary depth, with generic activation functions. As a special case, we then restrict to architectures with a single infinitely-wide layer, scalar outputs, and generalized cos-net activations. In this case, we show that BRG coarse-graining corresponds exactly to the momentum-shell ERG flow of a free scalar SFT. Our analytic results are corroborated by a numerical experiment in which an ensemble of asymptotically wide NNs are trained and subsequently renormalized using an information-shell BRG scheme.

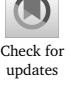
# 1 Introduction

There is a growing interest in obtaining theoretical descriptions of neural networks (NNs) in order to improve performance and address concerns of robustness, interpretability, and uncertainty estimation. Within computer science, a well-established approach to address these questions has been to apply the framework of Bayesian inference to NN training, commonly referred to as Bayesian Neural Networks (BNNs) [1–3]. This formalism provides a statistically well-grounded method to theoretically analyze NNs; in particular, it provides an avenue to assign uncertainty to NN predictions (for a pedagogical review see Refs. [1, 4]). Recently, alternative frameworks connecting NNs to statistical field theories (SFTs) have also been proposed as a way to theoretically analyze NNs [5–19]. Establishing this relation between NNs and SFTs (the NNFT correspondence) is intriguing as it opens up the possibility of leveraging decades of theoretical insights from the study of field theories in condensed matter and high-energy physics. For example, variants of the NNFT framework have been used to study critical phenomena and phase transitions in NNs [17, 20–25]. While many of these studies focus on NNs at initialization, it has also been posited that NN training admits a flow through the space of possible NNFTs [26, 27]. This suggestive language is intended to parallel the historical study of field theories within condensed matter and high-energy physics, in particular the exact renormalization group (ERG) [28, 29]. In this work, we draw on this rich history to provide a systematic, information-theoretic way to navigate the space of NNFTs. Schematically, viewing NNFTs through a Bayesian lens allows us to utilize recent developments connecting Bayesian inference to inverse ERG flows (Bayesian renormalization) [30, 31]. This perspective implies that each step of training corresponds to a point in the dual SFT space and these points are connected by an information-theoretic ERG flow.

The starting point of this joint framework, the NNFT correspondence, provides a many-to-one mapping from the space of NN architectures to the space of SFTs [12,32–34]. For a given choice of network architecture,[1] including its parameter distribution, this correspondence yields an exact field action for a dual SFT within the path integral formalism. Analogous to the Bayesian picture, a single NN output function can be viewed as a sample from its dual SFT. A sufficiently large, but finite, ensemble of NN output functions can be used to approximate the dual SFT, via the computation of arithmetic n-point functions. Extrapolating this line of thought, an infinite ensemble of NN output functions exactly reproduces the SFT dual, in the sense that the arithmetic n-point functions converge to those of the SFT. As a concrete example of this duality, infinitely-wide NNs with independent and identically distributed (i.i.d.) parameters correspond to free SFTs. Deviations from these architectural constraints will induce field interactions, whose coupling strengths are quantified by the departure from these limits (e.g. field couplings due to finite-width corrections scale as functions of inverse NN width [12,18]). Training a randomly initialized NN introduces statistical correlations among its parameters, violating the i.i.d. assumption [12]. Therefore, training an infinite ensemble of randomly initialized NNs approximates[2] a continuous flow among the dual SFTs, where field couplings change in response to NN parameter updates. The trajectories in the dual field space induced by training may seem ad-hoc from the SFT point of view, but the Bayesian perspective highlights their significance. In this picture, the NN ensemble at initialization is the prior distribution and every step of training yields a data-informed posterior. In other words, these flows among dual SFTs are generated via successive applications of Bayes' Theorem on the NN parameter distributions. Lastly, updated NN output functions may be processed through inverse Feynman rules [12,32–34] to explicitly compute the couplings and kinetic operators of the dual SFTs. In fact, all NN architectures have dual SFTs given explicitly in terms of actions in the path integral formalism. This naturally calls for a field-theoretic interpretation of the flows among the dual SFTs induced by training algorithms.

A quintessential method for creating flows in the space of SFTs is via renormalization group (RG) schemes.[3] In the picture proposed by Wilson [35,36], and refined by many others [28], SFTs can be connected via ERG equations which describe how the SFT changes with scale. Within physics contexts, many traditional SFTs have the property of spatial locality, thus establishing a natural scale with which to view these flows: energy (or, equivalently, distance). A traditional SFT viewed at low-energies (large distances) will generally be insensitive to high-energy (small distance) effects.[4] An ERG flow connects a traditional SFT at high-energies (UV theory) to its corresponding low-energy counterpart (IR theory) by systematically aggregating small-distance effects, i.e. coarse-graining. However, the SFT duals to NN architectures are most often spatially non-local; therefore, the ERG-like coarse graining of these NN models, and, by extension, their dual SFTs, must be organized according to a non-spatial scale. To this end, the authors in Refs. [30,31] have introduced a more general scale which lends itself to spatially non-local settings: information-theoretic similarity. In particular, they showed that ERG-like coarse-graining of NN parameters with respect to this scale amounts to inverting the process of Bayesian inference; thus, this procedure has been dubbed Bayesian RG (BRG).

---

[1]We note that typically the specification of 'architecture' includes a parameter distribution, however throughout this paper we want to highlight the importance of the parameter distributions in particular. Therefore, we will often distinguish the parameter distribution from the rest of the architecture.

[2]This approximation becomes exact in the limit of infinitesimal training time-steps.

[3]Here, we are broadly defining RG schemes to encompass both perturbative and non-perturbative approaches. While the latter is more general, we note that plainly saying "RG" colloquially refers to the former. Additional terms (i.e. "Wilsonian", "Exact", "functional", etc.) are often prepended to refer to the latter. To avoid confusion, in this work, we will exclusively be considering a very general, non-perturbative exact RG (ERG) scheme – Bayesian RG (BRG), which holds Wilsonian and Polchinski's ERG as special cases.

[4]More precisely, the high-energy effects can be captured by modifying the couplings between fields at low energies.

$$(\phi_\theta, \pi) \xrightarrow{\ NNFT\ } S[\phi]$$
$$\Big\downarrow{\scriptstyle BRG} \qquad\qquad \Big\downarrow{\scriptstyle BRG}$$
$$(\phi_\theta, \pi_\Lambda) \xrightarrow{\ NNFT\ } S_\Lambda[\phi]$$

Figure 1: Here, $\pi(\theta)$ is the parameter distribution of a NN, $\phi_\theta$, which has a dual SFT with action, $S[\phi]$. The BRG scheme coarse grains the original NN parameter distribution to $\pi_\Lambda$, indexed by the Fisher information scale, $\Lambda$. Updated NN ensembles have a transformed dual action, $S_\Lambda[\phi]$, connected to original action through an information-theoretic BRG flow.

On the other hand, recall that training a BNN is a step-by-step application of Bayesian inference. Utilizing the NNFT correspondence, we can interpret training as inducing an inverse BRG flow in SFT space from the information-theoretic 'IR' to a 'UV' fixed-point determined by the training data and objective. In certain circumstances, the BRG method becomes equivalent to traditional ERG schemes where the scale is based on spatial locality, e.g. Polchinski's ERG [37].

The combined BRG-NNFT framework maps each learning task to an information geometric flow in the dual SFT space. Given an ensemble of NNs trained with a specific learning objective, BRG can deduce the relative importance of each parameter in the model.[5] A particular information-theoretic 'cutoff' scale can be chosen to set a desired level of model distinguishability. Parameters falling below this scale (i.e. indistinguishable) can then be coarse grained. This can be done by setting them to 0 or by resampling them from the prior distribution defined at initialization [30]. The former can be used as a pruning strategy [30,38] while the latter is more intuitive from a field-theoretic perspective. A flow in NN architecture space is induced by incrementally increasing the cutoff and repeating this coarse-graining procedure. The NNFT correspondence then translates this to the dual SFT picture. The ensemble of trained NNs corresponds to a 'UV' field theory and the trajectory induced by BRG corresponds to a 'UV' → 'IR' flow that approximately inverts the 'IR' → 'UV' flow experienced during training. Combining these frameworks gives us a complete commutative diagram (see Figure 1) valid at any point along the flow.

This unified approach provides a pathway to present NN architectures in terms of dual SFTs, and the reverse. Establishing this connection could have many potentially interesting applications at the interface of physics and machine learning. For example, this correspondence potentially provides a method to construct representations of exotic SFTs so that they can be accurately and efficiently sampled. We note that this proposed application is highly related, although technically distinct, from existing literature [39–49] leveraging methods such as diffusion models [50,51] or normalizing flows [52] for the same tasks (see Section 5 for a more detailed discussion). As another example, this correspondence could be used to systematically explore the space of NNFTs and also potentially used as a new framework to study interesting phenomena in NN training behavior (e.g. criticality [20–22,25], grokking [53], feature learning [54–62], scaling laws [61,63–67]).

The current work sets the stage for these and other applications. In Section 2, we begin by reviewing Bayesian inference with the training of NN ensembles as a special case. Section 3 establishes the connection between the NNFT and BRG frameworks. In Section 3.1, we first review the NNFT correspondence as introduced in Refs. [12,32–34] and use this to connect

---

[5]Note that the BRG can be applied at any training step, including at initialization, however, since no training has occurred, the theory is already sitting at its 'IR' fixed point.

the pair $(\phi_\theta, \pi)$, representing a NN architecture and its parameter distribution, to its dual SFT with action, $S[\phi]$. Then, in Section 3.2, we review the BRG scheme [30, 31] for arbitrary inference models and discuss the similarities and differences between BRG and conventional Wilsonian ERG schemes. In Section 3.3, we then apply the BRG technique directly to NN ensembles. For a fixed architecture, BRG defines a family of parameter distributions indexed by the distinguishability scale, $\Lambda$. We then theoretically show that these parameter distributions can be passed through the NNFT correspondence to produce a family of SFTs with actions, $\{S_\Lambda[\phi]\}$, connected through an information-theoretic BRG flow. In Section 4, we analytically and numerically demonstrate the BRG-NNFT correspondence on analytically tractable cases. In Section 4.1, we explicitly compute the BRG flow for trained NNs with infinite width and arbitrary depth and activation functions. We find formulae for the renormalized mean (1-pt function) and covariance (2-pt function) which parameterize the NN as a renormalized Gaussian Process in this limit. In Section 4.1.1, we specialize these results to the case of a single layer NN with the generalized cos-net activation considered previously in Refs. [32, 33]. This allows us to take advantage of the fact that, with a well-chosen initialized $\pi(\theta)$, this architecture corresponds to a free scalar SFT having spectrum identical to conventional free scalar SFT. In this case, we show that BRG explicitly renormalizes the mass of the SFT, and can be interpreted in terms of a more familiar contraction of the momentum space of the SFT. Finally, in Section 4.2, we implement this joint scheme in a series of numerical experiments which corroborate our analytic results for asymptotically-wide NNs with ReLU [68] activations. We conclude with a discussion of our results in Section 5 where we also provide recommendations for future directions.

## 1.1   Related works

The primary contribution of the present work is to demonstrate the correspondence between the existing frameworks of NNFT [12, 32–34] and BRG [30, 31]. The joint BRG-NNFT is a general framework which connects to a many existing works in specific circumstances. Here, we briefly outline the rich landscape of existing literature and discuss relations to this work.

In this work, we will be using the NNFT correspondence described in Refs. [12, 32–34], which views a NN output function as a parameterization of a field, $\phi$, sampled from an SFT with action, $S[\phi]$. The NN itself is a mapping from $d$-dimensional Euclidean space to an output: $x \rightarrow \phi_\theta(x)$ with parameters $\theta$ and inputs $x \in \mathbb{R}^d$. However, there are many other works [5–19] which use alternate formulations to connect NNs with SFTs. For example, Refs. [9, 20] define a series of fields at each pre-activation, allowing them to make statements about the inter-layer interactions of these fields. There is also a rich literature, using methods from statistical physics to theoretically study many aspects of NN behavior [16, 17, 61, 62, 67, 69], as well as a closely related body of literature studying the theoretical behavior of Bayesian NNs (BNNs) [56, 59, 70–76]. As we will describe in subsequent sections, the present work connects directly to the training of BNNs. In particular, BRG-NNFT inherits the same computational difficulties as BNNs in estimating posterior distributions [77]. In this work, we consider a sample-estimate of posterior distributions by considering an ensemble of NNs. However, it would be interesting to explore BRG-NNFT in the context of other posterior-estimation schemes [1, 2, 77–80] in future work.

Many previous works [9,14,20,26,27,30,31,39,81–89] have also connected aspects of NNs to various RG schemes. In particular, various RG schemes have been used to study criticality and inter-layer information propagation [9,20,88], to understand NN generalization [14], and also to establish a direct relationship between Polchinski's ERG and the training of diffusion models [30, 31, 39, 89]. Furthermore, machine learning methods utilizing an information-theoretic RG scheme based on real space mutual information (RSMI) have also been used as

a way to construct physically relevant operators from data [82–85]. While these works also utilize a related information-theoretic quantity (mutual information), the relevance of operators are determined based on their long-distance behavior i.e. the cutoff scale is still based on a notion of spatial locality. Another series of works [26, 27] claims that the training of an ensemble of NNs can be interpreted as inducing an RG flow in the corresponding SFTs. To demonstrate this, they incrementally changed the weight distribution of their NN ensemble as a proxy for training, and showed that this induces an RG flow in the corresponding SFTs. In this work, we go a step further to actually train the NN ensemble explicitly, and then subsequently renormalize with BRG. Additionally, while both Refs. [26, 27] and our present analysis focus on the infinite-width GP limit of networks, we stress that the BRG-NNFT correspondence holds for any architecture. Our consideration of the infinite-width GP limit is precisely to demonstrate the BRG-NNFT correspondence in an analytically-tractable case that makes contact with existing results.

## 2 Preliminaries

In this section we provide an introduction to Bayesian inference for the purpose of performing and understanding machine learning. This section does not intend to provide a technical review of strategies for implementing Bayes[6] in the context of machine learning,[7] but rather a dictionary translating the mathematical tools and ideas from Bayes into those that are more familiar to readers of the machine learning literature. We begin by introducing Bayesian inference in the most general context of a parameterized probability model, and conclude by addressing how machine learning for NNs can be treated as a special case of such an approach. The familiar reader can safely skip this section and move to Section 3. For the reader who is in a rush, a summary of this section can be found in Table 1.

### 2.1 Introduction to Bayesian inference

Bayesian inference principally consists of two stages, model selection and learning. The model selection stage begins by understanding the structure of the system one is interested in learning about. In particular, we let $(S, d^d y)$ be a measure space in which a random variable, $Y$, takes its value. The stochastic properties of the variable $Y$ are encoded by specifying a probability distribution over $S$. A probability distribution is a measurable, normalized function $p \in L^1(S, d^d y)$ e.g.

$$\int_S d^d y \, p(y) = 1. \tag{1}$$

The probability that the random variable $Y$ takes value in a subset $\omega \subset S$ under the distribution $p$ is given by

$$\mathbb{P}_p(\omega) = \int_{\omega \subset S} d^d y \, p(y). \tag{2}$$

---

[6]In exposition, we will use the abbreviation 'Bayes' to refer to Bayesian inference.
[7]For those interested in such a survey, see e.g. [3].

Given a measurable function $f \in L^1(S, d^d y)$ we denote the expectation value of $f$ with respect to the distribution $p$ by[8]

$$\mathbb{E}_p(f) = \int_S d^d y \, p(y) \, f(y). \tag{4}$$

The premise of statistical inference is that we can learn about the distribution, $p_*$, of a random variable $Y$ by observing many realizations of $Y$. Hereafter we refer to $p_*$ as the *data generating* or *ground truth* distribution. A sample is a collection

$$D \equiv \{y_i\}_{i \in \mathcal{I}}, \tag{5}$$

where $\mathcal{I}$ is a countable index set and each $y_i$ is an independent draw from $p_*$. To infer on the distribution $p_*$ using the sample $D$ we postulate a class of models,

$$M \equiv \left\{ p_\theta \mid p_\theta \in L^1(S, d^d y), \int_S d^d y \, p_\theta(y) = 1 \right\}. \tag{6}$$

Each $p_\theta \in M$ is a candidate distribution for $Y$ parameterized by the elements $\theta$. Determining the set $M$ is referred to as *model selection*.

If the model class is well-selected it should be the case that the ground truth distribution $p_*$ belongs to the family $M$ and is realized at a unique point with parameters $\theta_*$, that is $p_* = p_{\theta_*}$. As a matter of probabilistic interpretation it is somewhat more natural to notate the distributions $p_\theta(y)$ by $p(y \mid \theta)$ with the understanding that $p_\theta(y)$ constitutes a probability model for $Y$ conditional on $\theta$ being the parameters of the true underlying distribution $p_*$. The space of models, $(M, d^n \theta)$, is itself a measure space and thus we may regard the parameters $\Theta$ as constituting a random variable taking values in this space. From this perspective, the aim of statistical inference is to determine a distribution $\pi \in L^1(M, d^n \theta)$ which encodes the probability that the ground truth distribution is located in any given region of model space:

$$\mathbb{P}_\pi(\omega \subset M) = \int_\omega d^n \theta \, \pi(\theta) = \text{Probability that } \theta_* \in \omega \subset M \text{ according to } \pi. \tag{7}$$

Once the sample $D$ has been collected, and the model space $M$ has been specified we are prepared to begin learning. The learning phase of Bayesian inference begins with the specification of a distribution over $M$ which we denote by $\pi_0$. The distribution $\pi_0$ is called the prior, and reflects the probability assigned to each region in model space before any information about the system has been incorporated. In this respect the prior is a reflection of our ignorance about the random variable $Y$. Learning in statistical inference describes the procedure by which the prior distribution is updated to reflect new information synthesized from the sample. In the Bayesian approach the update rule is provided by Bayes' law.

To motivate Bayes' law, let us consider a measure space with a Cartesian product structure $(S_1 \times S_2, d^{d_1} x_1 d^{d_2} x_2)$. An element $X = (X_1, X_2)$ taking values in $S_1 \times S_2$ may be regarded as a joint random variable and prescribed a joint probability distribution $p_{12}(x_1, x_2)$. From the joint distribution $p_{12}$ we can define two kinds of further probability distributions – marginal distributions and conditional distributions. The marginal distributions are given by

$$p_1(x_1) \equiv \int_{S_2} d^{d_2} x_2 \, p_{12}(x_1, x_2), \qquad p_2(x_2) \equiv \int_{S_1} d^{d_1} x_1 \, p_{12}(x_1, x_2), \tag{8}$$

---

[8]Notice that the probability measure can be written as the expectation value of an indicator function:

$$\mathbb{P}_p(\omega) = \mathbb{E}_p(\chi_\omega). \tag{3}$$

Here, $\chi_\omega(y)$ is the indicator function associated with the subset $\omega \in S$, i.e. it is equal to one if $y \in \omega$ and zero otherwise.

and correspond to probability distributions over $X_1$ ($X_2$) in which all of the randomness of $X_2$ ($X_1$) has been integrated out. With the marginal distributions (8) in hand we can define the conditional distributions

$$p_{1|2}(x_1 \mid x_2) \equiv \frac{p_{12}(x_1, x_2)}{p_2(x_2)}, \qquad p_{2|1}(x_2 \mid x_1) \equiv \frac{p_{12}(x_1, x_2)}{p_1(x_1)}. \tag{9}$$

One should regard $p_{1|2}(x_1 \mid x_2)$ ($p_{2|1}(x_2 \mid x_1)$) as a collection of probability distributions for $X_1$ ($X_2$) realized for each possible value that can be taken by the variable $X_2$ ($X_1$). In particular, $p_{1|2}(x_1 \mid x_2)$ ($p_{2|1}(x_2 \mid x_1)$) is the probability distribution for $X_1$ ($X_2$) conditional on $X_2 = x_2$ ($X_1 = x_1$). From (9) it is straightforward to see that

$$p_{12}(x_1, x_2) = p_1(x_1)p_{2|1}(x_2 \mid x_1) = p_2(x_2)p_{1|2}(x_1 \mid x_2), \tag{10}$$

this is Bayes' law.

Returning to the problem of statistical inference, to implement Bayes' law we should regard $(D, \Theta)$ as defining a joint random variable on the measure space $S^{\times |D|} \times M$.[9] Here we have regarded the sample $D = \{y_1, \ldots, y_{|D|}\}$ as a collection of independent and identically distributed random variables taking values in $S$. The distribution for $D$ conditional on $\theta$ is given by

$$\mathcal{L}(D \mid \theta) \equiv \prod_{i \in \mathcal{I}} p(y_i \mid \theta). \tag{11}$$

The object defined in (11) is called the *Likelihood* since it encodes the probability of observing a particular sample $D$ under the assumption that the data generating distribution is parameterized by $\theta$. The marginal probability distribution for $\theta$ is given by $\pi_0$ as the beginning of training, and thus the joint distribution over sample and parameters is

$$p_{D,\Theta}(D, \theta) = \pi_0(\theta)\mathcal{L}(D \mid \theta). \tag{12}$$

Using (8) we can therefore obtain the marginal distribution for the sample by integrating out $\theta$

$$Z(D) \equiv \int_M d^n\theta \, \pi_0(\theta)\mathcal{L}(D \mid \theta). \tag{13}$$

Then, finally, we can use (9) to define a new distribution over parameters conditional on the observed data $D$:

$$\pi(\theta \mid D) \equiv \frac{1}{Z(D)}\pi_0(\theta)\mathcal{L}(D \mid \theta). \tag{14}$$

One can clearly see that (14) is nothing but an application of Bayes' law (10). In the following we will sometimes denote $\pi(\theta \mid D)$ by $\pi_D(\theta)$.

The distribution $\pi_D$ is called the *posterior* and has the following interpretation; it is a probability weighting over the space of possible models informed by the collection of a given sample of data. Given the distribution $\pi_D$ we can also construct the so-called *posterior predictive* distribution

$$p(y \mid D) \equiv \mathbb{E}_{\pi_D}\Big(p(y \mid \Theta)\Big) = \int_M d^n\theta \, \pi_D(\theta)p(y \mid \theta). \tag{15}$$

As the notation suggests, the posterior predictive can be interpreted as the "expected value" over the model class (6) with respect to the weighting defined by the posterior. In summary, Bayesian inference can be regarded as the arrow

$$\pi_0(\theta) \mapsto \pi_D(\theta), \tag{16}$$

---

[9]The measure on this space is a tensor product of $|D|$ copies of the measure on $S$ with the measure on $M$ e.g. $d^d y_1 \cdots d^d y_{|D|} d^n\theta$.

in which the distribution over model space is trained around the data generating model via the observation of data.

To clarify some of the abstract concepts introduced in this section, let us consider a standard implementation of Bayesian inference. Suppose we are interested in determining the probability of heads for a possibly unfair coin. To perform inference in this example we collect a sample of $N$ tosses of the coin, $D = \{y_i\}_{i=1}^{N}$, where each $y_i$ belongs to the sample space $S = \{H, T\}$ and $H$, $T$ indicates the outcome of 'head' and 'tail', respectively. The data generating distribution of the coin is Bernoulli with probability of heads given by some $\theta_* \in [0, 1]$. In particular, $p_*(y) = \text{Bern}_{\theta_*}(y)$ where here

$$\text{Bern}_\theta(y = H) = \theta, \qquad \text{Bern}_\theta(y = T) = 1 - \theta. \tag{17}$$

The natural model class for this inference problem is the space of Bernoulli distributions

$$M = \{\text{Bern}_\theta \mid \theta \in [0, 1]\} \simeq [0, 1]. \tag{18}$$

This model space is equivalent to the interval $[0, 1]$, which corresponds to the possible probabilities which the coin could assign to a head.

To perform Bayesian inference we must choose a prior probability on the space of models. For simplicity, let us take the flat prior,

$$\pi_0(\theta) = 1, \qquad \theta \in [0, 1]. \tag{19}$$

This ensures that the probability that $\theta$ is contained in any subinterval of the model space is simply equal to the size of the interval. Intuitively, the flat prior places equal probability to all of the possible values of the parameter $\theta$. The likelihood in this example is equal to the probability of observing the given sample under the assumption that the coin has a probability $\theta$ of landing on heads. Specifically,

$$\mathcal{L}(D \mid \theta) = \prod_{i=1}^{N} \text{Bern}_\theta(y = y_i) = \theta^{N_H}(1 - \theta)^{N_T}, \tag{20}$$

where $N_H$ ($N_T$) is the number of heads (tails) observed in the sample. The joint probability density for the sample and the parameter $\theta$ is given by

$$p_{D,\theta}(D, \theta) = \theta^{N_H}(1 - \theta)^{N_T}. \tag{21}$$

Integrating this quantity over $\theta$, and applying Bayes' law we obtain the normalized posterior

$$\pi_D(\theta) = \frac{(N_H + N_T + 1)!}{N_H! N_T!} \theta^{N_H}(1 - \theta)^{N_T} = \text{Beta}_{N_H+1, N_T+1}(\theta). \tag{22}$$

This is a Beta distribution with parameters $\alpha = N_H + 1$ and $\beta = N_T + 1$.

Such a distribution has a mean equal to $\frac{N_H + 1}{N + 2}$, which asymptotes to the true underlying parameter for sufficiently large samples. For finite samples, (22) is essentially centered[10] on the empirical probability estimate. In this way, the Bayesian posterior reproduces the maximum likelihood estimate, but also allows for a direct computation of the probability that the true underlying parameter belongs to any interval in the model space. This allows, among other things, for the formation of probabilistic confidence intervals.[11]

---

[10]The reason the posterior mean is not equal to $N_H/N$ is due to the effect of the prior. A flat prior on this model space can be interpreted as a Beta distribution with parameters $\alpha = \beta = 1$. This has the quantitative effect of adding two additional samples; one head and one tail.

[11]It's worth noting that the variance of the posterior distribution is given by

$$\text{Var}(\theta \mid D) = \frac{(N_H + 1)(N_T + 1)}{(N + 2)^2(N + 3)}, \tag{23}$$

which scales as $1/N$ for large $N$. This means that the width of the posterior will becomes sharper around the maximum likelihood estimate as the sample becomes larger.

## 2.2 Bayesian inference for neural networks

Having presented an overview of Bayesian inference for general parameterized probability models we can now discuss inference for NNs as a special case. In this subsection we will take the point of view that a NN is nothing but a very flexible model for approximating general functions. Thus, the problem of inference for NNs is one of determining a distribution over a function valued random variable.

To be precise, a NN is a map

$$\phi : P \to \text{Maps}(I, O). \tag{24}$$

Here, $P$ is the space of parameters for the NN, and $\text{Maps}(I, O)$ is the set of maps from an input space $I$ to an output space $O$. One may alternatively regard the NN as a map

$$\phi : P \times I \to O. \tag{25}$$

We will intermittently use the notations $\phi_\theta(x)$, and $\phi(x \mid \theta)$ to stress these two compatible pictures. Implicit to the specification of (24) is the selection of a particular set of parameters on which the NN depends, and the functional form of that dependence – i.e. the choice of architecture.

Once the NN architecture, including its initialization distribution $\pi_0$, has been specified one can begin learning. The training data is presented as a set of matched pairs, $D \equiv \{x_i, \phi_i^*\}_{i \in \mathcal{I}}$ where $x_i \in I$ and $\phi_i^* \in O$. Here, $\phi_i^*$ is the value of the target function $\phi^*$ which we would like our NN to approximate evaluated at $x_i$, i.e. $\phi_i^* = \phi^*(x_i)$. Typically, training a NN is understood as a problem in convex optimization whereupon the network is asked to tune its parameters in order to minimize a prescribed loss function $L(D \mid \theta)$. To obtain robust parameter estimates, one performs training on an ensemble of networks with values initialized from the distribution $\pi_0$. The resulting initialized models are subsequently trained using a method of choice, such as gradient descent. The result of training is to acquire a set of parameter estimates, one for each network in the ensemble, which we denote by $\{\theta_n(D)\}_{n=1}^K$. One may interpret $\{\theta_n(D)\}_{n=1}^K$ as draws from some new parameter distribution identifiable with a Bayesian posterior, $\pi_D$. Thus, in this sense, network training can be interepted according to the same arrow (16) which characterizes Bayesian inference. These estimates are subsequently pooled to arrive at an optimal $\theta_*(D)$ resulting in the trained network $\phi_{\theta_*(D)}$.

Let us now turn our attention to reinterpreting the procedure for NN learning as Bayesian inference. The first step is to frame the problem of learning a function in a probabilistic language. In particular, we regard $(\text{Maps}(I, O), D\phi)$ as a measure space, and treat $\phi \in \text{Maps}(I, O)$ as a random variable on this space with distribution $p(\phi)$.[12] It is instructive to tease out the meaning of the measure $D\phi$ and the distribution $p(\phi)$ in the following way. First, let $C = \{x_i\}_{i=1}^N$ be a mesh on the input space – that is a countable collection of points. Evaluated on the mesh $C$ a function $\phi \in \text{Maps}(I, O)$ can be regarded as a collection of $N$ elements with values in $O$. We introduce the notation

$$\phi_C \equiv \{\phi(x_i)\}_{i=1}^N, \tag{26}$$

to refer to the function evaluated on this mesh. The element (26) may therefore be regarded as an $N$-dimensional joint random variable taking values in $O$. When considering only points in the mesh, the space $(\text{Maps}(I, O), D\phi)$ is therefore reduced to $O^{\times N}$ with the typical product

---

[12]For the usual reasons it is risky to regard $D\phi$ as a genuine Lebesgue measure. Nevertheless, in the examples we have in mind the space $\text{Maps}(I, O)$ can be discretized leading to a formal definition of the measure, as described in this section.

measure. Thus, the functional probability distribution $p(\phi)$ induces a distribution $p_C(\phi)$ such that

$$\mathbb{P}_{p_C}(\omega) = \int_{\omega \in O^{\times N}} d\phi(x_1) \cdots d\phi(x_N) \, p_C(\phi(x_1), \dots, \phi(x_N)), \qquad (27)$$

can be interpreted as the probability that $\{\phi(x_i)\}_{i=1}^N$ takes values in $\omega \subset O^{\times N}$. The measure $D\phi$ and the distribution $p(\phi)$ are defined by formally taking a limit so that the mesh covers every point in the input space. In this sense, one may regard a function valued random variable as a collection of jointly distributed random variables taking values in the output space, with one random variable for each point $x \in I$.

Having understood the sense in which $\phi$ can be treated as a random variable taking values in the space $\text{Maps}(I, O)$ one could perform Bayesian inference over $\phi$ simply by following the steps outlined in Section 2. To reproduce the analysis we have presented, we simply recognize that the specification of a model architecture (24) automatically induces a parameterized probability model over $\phi$. In the strict sense, conditional on $\theta$ the random variable $\phi$ is simply set equal to $\phi_\theta$. In other words, we have[13]

$$p(\phi \mid \theta) \equiv \delta(\phi - \phi_\theta). \qquad (30)$$

The initialization distribution $\pi_0$ now plays the role of a Bayesian prior. Given the data $D = \{(x_i, \phi_i^*)\}_{i \in \mathcal{I}}$ one can update the prior using Bayes' law to obtain the distribution $\pi_D(\theta)$. From this perspective, rather than having an ensemble of networks we have promoted the parameters of network to the status of random variables. The gradient descent estimates $\{\theta_n(D)\}_{n=1}^K$ can now be regarded simply as draws from the posterior distribution, $\pi_D(\theta)$, and the optimal estimate is obtained by taking the maximum a posteriori estimate

$$\theta_*(D) \equiv \text{argmax}_{\theta \in P} \, \pi_D(\theta). \qquad (31)$$

More to the point, the 'state' of the network is described entirely by the distribution $\pi_D$. The resulting probability distribution over functions is given by the posterior predictive

$$p_D(\phi) \equiv \int_P d^n\theta \, \pi_D(\theta) p(\phi \mid \theta). \qquad (32)$$

In the following sections we will discuss the sense in which the distribution $p_D(\phi)$ can be interpreted as a field theory, and we will provide a systematic approach for evaluating the integral (32) as a perturbation series in sampled cumulants.

### 2.3 Summary

For the convenience of the reader we present here a summary of the correspondence between Bayesian inference and machine learning:

---

[13]In practice it is advisable to modify (30) so that the distribution has some finite width e.g.,

$$p(\phi \mid \theta, \Sigma) \equiv \mathcal{N}(\phi \mid \phi_\theta, \Sigma). \qquad (28)$$

Here, $\mathcal{N}(y \mid \mu, \Sigma)$ is the normal density with mean parameter $\mu$ and covariance parameter $\Sigma$. (30) can be regarded as a limit of (28) in which the covariance goes to zero. More generally, one can take the distribution over network functions given parameters to be of the form

$$p(\phi \mid \theta) \sim e^{-L(\phi \mid \theta)}, \qquad (29)$$

where here $L(\phi \mid \theta)$ is the chosen training loss and the proportionality is up to normalization. In this case minimizing the loss is rendered exactly equivalent to maximum likelihood estimation. For example, the normal likelihood (30) can be regarded as the probability model associated with $L^2$ losses.

Table 1: Summary of the relationship between Bayesian inference and machine learning.

| Bayesian Inference | Machine Learning |
| --- | --- |
| Random variable, $Y \in S$ | Predictive function, $\phi \in C^{\infty}(I, O)$ |
| Data generating distribution, $p_*$ | Target function, $\phi^*$ |
| Model class, $p(y \mid \theta)$ | Network architecture, $\phi_\theta$ |
| Sample, $D = \{y_i\}_{i \in \mathcal{I}}$ | Training data, $D = \{x_i, \phi_i^*\}_{i \in \mathcal{I}}$ |
| Prior, $\pi_0(\theta)$ | Initialization distribution, $\pi_0(\theta)$ |
| Learning via Bayes' theorem | Training via loss minimization |
| Posterior, $\pi_D(\theta)$ | Ensemble of parameter estimates, $\{\theta_n(D)\}_{n=1}^{K}$ |

$$
\begin{array}{ccc}
(\phi_\theta, \pi) & \xrightarrow{NNFT} & S[\phi] \\
\downarrow{\scriptstyle BRG} & & \downarrow{\scriptstyle BRG} \\
(\phi_\theta, \pi_\Lambda) & \xrightarrow{NNFT} & S_\Lambda[\phi]
\end{array}
$$

Figure 2: A commutative diagram which unifies NNFT and BRG. The composition of these maps *defines* a new proposal for constructing information geometric flows through the space of field theories. We interpret these BRG flows as a suitable replacement for standard field theoretic ERG flows in general contexts where spatially local coarse graining is not sufficient for implementing a meaningful flow.

# 3  NNFT + BRG

As we have alluded to in the introduction, the story of this work can largely be told through the commutative diagram shown in Figure 2 (the diagram itself is a repeat of that in Figure 1 but the caption adds additional context to summarize the conceptual results of Section 3).

Each of the three subsections of this section coincide with one leg of the diagram (Figure 2). In Section 3.1, we introduce the Neural Network Field Theory (NNFT) correspondence, which is a tool for translating network architectures with stochastic parameters into explicit statistical field theories (SFTs). In Section 3.2, we establish the set up for Bayesian Renormalization, which is an information geometry inspired approach to coarse graining over the space of models in a statistical inference experiment. Finally, in Section 3.3, we combine NNFT and BRG to obtain an information theoretic renormalization group for exploring the space of SFTs.

## 3.1  Neural network-field theory correspondence

In [12, 32, 33], it has been demonstrated how one can map a NN with prescribed architecture and a given distribution over parameters into a continuum SFT. As we have already discussed, from the probabilistic point of view a NN is a function valued random variable $\phi \in \text{Maps}(I, O)$,

$$
(\phi_\theta, \pi) \xrightarrow{NNFT} S[\phi]
$$

Figure 3: NNFT maps a fixed NN architecture, $\phi_\theta$, with parameter distribution, $\pi(\theta)$, into a Euclidean field theory with action, $S[\phi]$.

and thus a field theory for $\phi$ would simply be a probability law $p(\phi)$.[14] Formally, such a distribution can be realized as follows – we regard $(\phi, \theta)$ as a pair of joint random variables corresponding to the realized network and its parameters. Given a fixed architecture, the distribution for $\phi$ conditional on $\theta$ is (30) and thus, given a marginal distribution over parameters $\pi$ we can simply write

$$p(\phi) = \int_P d^n\theta \; \pi(\theta) \, \delta(\phi - \phi_\theta). \tag{33}$$

As a rule, however, (33) is a highly intractable integral.

Instead of attempting to compute (33) we can exploit the fact that it defines a duality between the parameter space and the field space. That is, given any functional $F(\phi)$ we can compute its expectation value either as a functional integral

$$\mathbb{E}_p(F) = \int D\phi \; p(\phi) F(\phi), \tag{34}$$

or, using (33), as an expectation value over the parameter distribution

$$\mathbb{E}_p(F) = \int D\phi \int_P d^n\theta \; \pi(\theta) \, \delta(\phi - \phi_\theta) F(\phi) = \int_P d^n\theta \; \pi(\theta) \, F(\phi_\theta) = \mathbb{E}_\pi(F(\phi_\theta)). \tag{35}$$

As a specific instance of (35) we can compute the moment generating functional (i.e. the partition functional)

$$Z[J] = \mathbb{E}_\pi(e^{\int_I d^d x \, J(x)\phi_\theta(x)}) = \int_P d^n\theta \; \pi(\theta) e^{\int_I d^d x \, J(x)\phi_\theta(x)}, \tag{36}$$

here $J(x)$ is a source function on the input space. From (36) we obtain the statistical moments of $p$

$$G^{(n)}(x_1, \ldots, x_n) \equiv \frac{\delta^n}{\delta J(x_1) \cdots \delta J(x_n)} Z[J] \bigg|_{J=0} = \mathbb{E}_\pi \bigg( \phi_\theta(x_1) \cdots \phi_\theta(x_n) \bigg). \tag{37}$$

In practice, one can numerically approximate (37) via sampling. That is, one may sample $T$ draws of parameters from the distribution $\pi$, $\{\theta_k\}_{k=1}^T$, which correspond to $T$ network realizations $\{\phi_{\theta_k}\}_{k=1}^T$. Given the sample of field configurations, one can compute sample statistics for the distribution $p(\phi)$. In particular,

$$\tilde{G}^{(n)}(x_1, \ldots, x_n) \equiv \frac{1}{T} \sum_{k=1}^T \prod_{j=1}^n \phi_{\theta_k}(x_j), \tag{38}$$

where the tilde indicates that this is now the sampled estimate of the $n$-point correlator for the distribution $p$; with an accuracy roughly proportional to the size of $T$.

Given the data of $G^{(n)}(x_1, \ldots, x_n)$ (or appropriate sample versions) we can evaluate the integral (33) using the Edgeworth expansion [92, 93]. Let $p_0(\phi)$ be a Gaussian random process with mean and covariance set by the first and second sampled cumulants derived from (38). In other words, $p_0(\phi) \sim e^{-S_{\text{free}}(\phi)}$ is the best free field approximation for the theory $p$. Then, we can express the distribution $p$ in terms of the series expansion

$$p(\phi) = p_0(\phi) + \sum_{n=3}^\infty \frac{(-1)^n}{n!} \int_{I^{\times n}} \prod_{i=1}^n d^d x_i \; G^{(n)}(x_1, \ldots, x_n) \frac{\delta^n p_0(\phi)}{\delta\phi(x_1) \cdots \delta\phi(x_n)}. \tag{39}$$

---

[14]In this sense we are implicitly working in the context of Euclidean or statistical field theory. As always, one can traverse from the statistical field theory $p$ to a bona-fide quantum field theory provided the Osterwalder-Schrader axioms are met [33], see also [90, 91].

Table 2: Summary of NNFT correspondence.

| Neural Network | Field Theory |
|---|---|
| Inputs $x$ | Euclidean space (or momentum) |
| Outputs $f$ | Field $\phi$ |
| Cumulants $G_c^{(n)}(x_1, \ldots, x_p)$ | Connected correlators $G_c^{(n)}(x_1, \ldots, x_p)$ |
| Kernel | $G^{(2)}(x_1, x_2)$ |
| Log-likelihood | Action $S[\phi]$ |

Alternatively, (39) can be written in the form

$$p(\phi) = \frac{1}{Z[0]} \exp\left( \sum_{n=3}^{\infty} \frac{(-1)^n}{n!} \int \prod_{j=1}^{n} d^d x_i \; G_c^{(n)}(x_1, \ldots, x_n) \frac{\delta}{\delta\phi(x_1)} \cdots \frac{\delta}{\delta\phi(x_n)} \right) e^{-S_{\text{free}}(\phi)}, \quad (40)$$

here $G_c^{(n)}(x_1, \ldots, x_n)$ are the connected correlation functions (cumulants) obtained via the moment-cumulant formula from $G^{(n)}(x_1, \ldots, x_n)$. At this point, we should stress that the correspondence (40) is valid for any NN architecture, with any parameter distribution, without restriction to infinite width or initialization. One can compute (40) as long as one can obtain high-precision sampled cumulants $\tilde{G}_c^{(n)}(x_1, \ldots, x_n)$.

In [32] it has been shown that $p(\phi) \propto e^{-S[\phi]}$ has the form of a generically interacting Euclidean field theory with action $S[\phi] = S_{\text{free}}(\phi) + S_{\text{int}}(\phi)$ given by

$$S_{\text{free}}(\phi) = \int d^d x_1 \, d^d x_2 \; \phi(x_1) G_c^{(2)}(x_1, x_2)^{-1} \phi(x_2), \quad (41)$$

$$S_{\text{int}}(\phi) = \sum_{n=3}^{\infty} \int \prod_{j=1}^{n} d^d x_i \; g^{(n)}(x_1, \ldots, x_n) \phi(x_1) \cdots \phi(x_n). \quad (42)$$

In (42), $g^{(n)}(x_1, \ldots, x_n)$ are space dependent couplings obtained from the connected correlation functions (cumulants) via a set of 'inverse Feynman rules' [32]. The free action is well-defined for models with an invertible, i.e. has full rank, 2-pt function, with $G_c^{(2)}(x, y)^{-1}$ regarded as the inverse of the two point function or the network kernel. Thus, there is a direct correspondence between a generic NN and an SFT with Euclidean action $S[\phi]$. This correspondence, introduced in Refs. [12, 34], is summarized in Table 2.

The standard result of Neal that an infinitely wide NN with independent and identically distributed parameters can be regarded as a Gaussian Random Process [3] is automatically encoded in the NNFT correspondence. It follows from the central limit theorem (CLT) which guarantees that all of the connected correlation functions of such a network, $G_c^{(n)}$, are zero for $n > 2$. Thus, following (41), we see that $p(\phi) = \frac{1}{Z[0]} e^{-S_{\text{free}}(\phi)}$ coincides with a free field theory in this case. The power of NNFT truly presents itself when considering perturbations away from Neal's seminal result, which may now be interpreted through the lens of perturbations away from a free field theory [32]. From this perspective, a natural way to generate non-Gaussianities (i.e. interactions) in an NNFT is to break the assumptions which are required for the central limit theorem to hold. Two straightforward ways of doing so are to move away from the strict $N \to \infty$ limit, or to choose a distribution $\pi$ in which parameters are no longer independent. The couplings appearing in $S_{\text{int}}$ (42) will have orders of magnitude associated with the extent to which the CLT is violated. For example, in the case that interactions arise from the finite width of the network while retaining the i.i.d. assumption over parameters it can be shown that the connected correlation functions scale as

$$G_c^{(n)}(x_1, \ldots, x_n) \propto \frac{1}{N^{n/2-1}}. \quad (43)$$

The couplings, $g^{(r)}(x_1, \ldots, x_r)$, can, in turn, be expressed as a sum of all connected $r$-point Feynman diagrams which are constructed with $G_c^{(n)}$ vertices for all possible $n$. For example, in the case of i.i.d. initialized networks, the dependence of $g^{(4)}(x_1, \ldots, x_4)$ on $G_c^{(4)}$ results in $g^{(4)}$ scaling as $1/N$ at leading order, while other connected correlation functions contribute to subleading effects at $O(1/N^2)$ or less. More details on the Feynman rule prescription for general $g^{(r)}$, including the cases of non-i.i.d. initializations, can be found in Section 3.2 and 3.3 of Ref. [32].

Notice in all of our discussions that the distribution $p$ (40) depends explicitly on the parameter distribution $\pi$ due to the field space/parameter space duality. Thus, for each parameter distribution $\pi$ we obtain a different field theory. From the Bayesian perspective, training corresponds to promoting the prior or initialization distribution $\pi_0$ to the posterior or trained distribution $\pi_D$. From the NNFT perspective this should be interpreted as generating a flow in the space of field theories from $p_0$ – the NNFT generated by the prior weight $\pi_0$ – to $p_D$ – the NNFT generated by the posterior weight $\pi_D$.

The distribution $p_0$ associated with the prior is untrained and thus mostly ignorant to the distinguished features of the target data. By contrast, the field theory $\pi_D$ should be sharply attuned to the features of the target. In this respect we might be inclined to regard the process of training as inducing a flow from the 'IR' to the 'UV' in the space of NNFTs. Here, we are borrowing terminology from physics but implementing this language in a slightly nuanced way. By IR we mean an effective theory which is based on a limited amount of information about the underlying sample. By UV we mean a more complete theory trained on a large amount of information about the underlying sample. This information theoretic interpretation of IR and UV theories has been explicated on in [30, 31].

As a final note, we have, up to this point, mostly emphasized NNFT as a tool for deriving field theory representations of parameter distributions which are obtained in a NN training setting. However, we can also utilize NNFT in the reverse direction – e.g. to *design* NN architectures along with parameter distributions that correspond to an SFT of our choice. One approach to accomplishing this proceeds as follows: We begin with a single, infinite width hidden layer NN with i.i.d. parameter distributions $\pi_{\mathrm{GP}}$ chosen to generate the target free action $S_{\mathrm{free}}(\phi)$. We then perturb the free field theory by introducing our desired interactions in the NNFT path integral, e.g. by adding terms of the form $\mathcal{O}(\phi)$ into the action. In practice, however, such deformations of action are generated via statistical correlations of the form $\exp(-\mathcal{O}(\phi_\theta))$ induced among NN parameters through the dual architecture space. After each such deformation, the partition functions of both field and architecture spaces can be written as

$$Z[J] = \int D\phi \, e^{-(S_{\mathrm{free}}(\phi) + \mathcal{O}(\phi)) + \int_I d^d x J(x)\phi(x)} = \int_P d^n\theta \, \pi_{\mathrm{GP}}(\theta) e^{-\mathcal{O}(\phi_\theta)} \, e^{\int_I d^d x J(x)\phi_\theta(x)}, \quad (44)$$

identifying a new architecture having a non-i.i.d parameter distribution

$$\pi_{\mathcal{O}}(\theta) := \pi_{\mathrm{GP}}(\theta) e^{-\mathcal{O}(\phi_\theta)}.$$

We interpret the pair $(\phi_\theta, \pi_{\mathcal{O}})$ as the transformed architecture whose dual field theory has a perturbative interaction $\mathcal{O}(\phi)$. For explicit implementations of this approach we refer the reader to [32, 33].

## 3.2 An information theoretic form of Wilsonian renormalization

In light of the previous section, training a NN might be regarded as inducing a flow from the information-theoretic IR to the UV in the space of NNFTs. In this section, we would like to describe an approach to defining information theoretic RG flows in the space of NNFTs from

$$(\phi_\theta, \pi) \xrightarrow{\text{BRG}} (\phi_\theta, \pi_\Lambda)$$

Figure 4: BRG is an information geometric coarse graining scheme that realizes a family of parameter distributions $\pi_\Lambda$ indexed by a distinguishability scale $\Lambda$.

the UV to the IR. It is tempting to simply apply the machinery of standard renormalization directly to NNFTs, however this assumes properties of the NNFT which are rarely satisfied in practice, such as locality.

On the other hand, the more general Bayesian Renormalization [30] is an information theoretic renormalization scheme that inverts a Bayesian learning experiment by systematically discarding information in a hierarchy of scales defined by the Fisher information metric. On the learning side, a Bayesian RG starts with a trained posterior, $\pi_{UV}(\theta)$, and produces a highly diffuse distribution, $\pi_{IR}(\theta)$. Through the NNFT correspondence, this will equivalently induce a flow from a trained NNFT, $p_{UV}(\phi)$, to a diffuse effective theory, $p_{IR}(\phi)$. For a complete introduction to Bayesian RG we refer the reader to [30, 31, 94]. In this section, our goal will be to motivate an information theoretic/geometric interpretation of the Bayesian RG scheme and understand the sense in which it generalizes more familiar notions of renormalization that have their origin in field theory.

To begin, let us recall some notions from information geometry [95, 96]. As in Section 2, our starting point is a measure space $(S, d^d y)$ in which a random variable $Y$ takes values. We moreover have $(M, d^n \theta)$ the space of models, which corresponds to a space of parameterized probability distributions for $Y$. Restricting our attention to finitely parameterized models the map

$$p : \mathbb{R}^n \to M, \ \theta \mapsto p_\theta, \tag{45}$$

can be regarded as a local coordinate system endowing $M$ with the structure of a differentiable manifold.

The space $M$ can moreover be seen to possess a canonical Riemannian structure in the following sense. Firstly, we can define the tangent bundle of $M$ at the point $\theta$ as the set of measurable functions $f \in L^1(S, d^d y)$ whose expectation value vanishes at $\theta$:

$$T_\theta M \equiv \{ f \in L^1(S, d^d y) \mid \mathbb{E}_{p_\theta}(f) = \int_S d^d y \, p_\theta(y) f(y) = 0 \}. \tag{46}$$

Geometrically, (46) is a good definition for the tangent bundle because such functions correspond to the set of legal deformations which preserve the normalization of the probability measure $p_\theta$. That is if $p_\theta \mapsto p_\theta(1 + f)$, the resulting measurable function will only be integral normalized if $f$ has zero expectation value. The tangent bundle (46) is spanned by the so-called *score vectors*

$$\ell_A \equiv \frac{\partial \ln p_\theta(y)}{\partial \theta^A}. \tag{47}$$

The expectation value of (47) is zero for each $A$ precisely by making use of the normalization condition

$$\mathbb{E}_{p_\theta}(\ell_A) = \frac{\partial}{\partial \theta^A} \left( \int_S d^d y \, p_\theta(y) \right) = 0. \tag{48}$$

As defined, the tangent bundle admits a natural symmetric, positive definite, bilinear in the form of the covariance

$$\mathcal{I} : T_\theta M^{\times 2} \to C^\infty(M), \quad \mathcal{I}(f, g) \equiv \mathbb{E}_{p_\theta}(f g). \tag{49}$$

One may therefore regard (49) as defining a Riemannian metric on $M$. Working in the basis (47) the metric (49) has components

$$\mathcal{I}_{AB}(\theta) \equiv \mathbb{E}_{p_\theta}\left(\ell_A \ell_B\right) = \int_S d^d y \, p_\theta(y) \frac{\partial \ln p_\theta(y)}{\partial \theta^A} \frac{\partial \ln p_\theta(y)}{\partial \theta^B}, \tag{50}$$

which are the elements of the *Fisher information matrix*. Alternatively, the Fisher matrix can be obtained by performing a quadratic expansion of the relative entropy

$$D_{KL}(p_\theta \parallel p_{\theta'}) \equiv \mathbb{E}_{p_\theta}\left(\ln p_\theta - \ln p_{\theta'}\right). \tag{51}$$

Taking $\theta' = \theta + \delta\theta$ and expanding, we have

$$D_{KL}(p_\theta \parallel p_{\theta+\delta\theta}) = \frac{1}{2}\mathcal{I}_{AB}\delta\theta^A\delta\theta^B + \mathcal{O}(\parallel \delta\theta \parallel^3). \tag{52}$$

In this respect, we interpret (50) as an infinitesimal measure of the distinguishability between probability models for the random variable $Y$.

From a statistical inference perspective, the Fisher metric plays a rather central role. In this paper, we will concentrate on the fact that the Fisher metric provides an automatic quantification of the relevance of parameters appearing in given model [97]. In particular, the diagonal elements[15] of the Fisher matrix, $\lambda_A \equiv \mathcal{I}_{AA}(\theta)$, may be regarded as quantifying the sensitivity of the $A^{th}$ parameter to the acquisition of new data conditional on the data generating model having parameter estimate $\theta$. In this context, the following terminology is often utilized to categorize parameters: We say that the $A^{th}$ parameter is more *stiff* than the $B^{th}$ parameter if $\lambda_A/\lambda_B > 1$. Conversely, if $\lambda_A/\lambda_B < 1$, we say that the $A^{th}$ parameter is more *sloppy* than the $B^{th}$ parameter.

In general, a stiff parameter is one which can be reliably fit with a reasonable amount of data. By contrast, sloppy parameters tend to require extremely large or precise data in order to fit. For this reason, a model which possesses a large number of sloppy parameters retains a high degree of degeneracy in the sense that these parameters may be varied largely without impacting the performance of the model dramatically.[16] Information geometrically, the diagonal element $\lambda_A$ is a proxy for the radius of the model space in the $A^{th}$ parameter direction. Thus, one may qualitatively regard the model space as consisting of some number of "macroscopic" dimensions corresponding to the stiff parameters, and some number of highly compact dimensions corresponding to the sloppy ones.

We would now like to present a comparison between the observations of the previous paragraph and Wilson's approach to renormalization [35,36]. Given a local quantum field theory, Wilson's idea was to organize the field degrees of freedom according to a Fourier decomposition of the relevant Laplace operator. Of course, these Fourier modes simply correspond to the momenta of each mode. For simplicity we will work in the Euclidean signature so that the field theory may be regarded as a probability theory for the field, $\Phi$, viewed as a function valued random variable. Then, Wilson's picture can be contextualized as parameterizing the distribution over $\Phi$ in terms of the momenta $p$. Due to experimental limitations, only the effective low energy features of any given field theory can be matched to data. Thus, a continuum

---

[15]Strictly speaking it would be more geometrically meaningful to diagonalize the Fisher metric so that the following statements are basis independent. With that being said, diagonalization is very computationally costly and in a model with a sparse Fisher metric the diagonal components in a chosen basis are a good proxy for the eigenvalues [95].

[16]That is, large changes in the actual sloppy parameter value $\theta^A$ correspond to comparatively short distances as measured by the information metric, and in turn leave the model effectively unchanged in the relative entropy sense.

$$(\phi_\theta, \pi_\Lambda) \xrightarrow{NNFT} S_\Lambda[\phi]$$

Figure 5: Applying NNFT to a family of Bayesian renormalized parameter distributions realizes a one parameter family of Euclidean field theories which we interpret as an information theoretically coarse grained flow in the space of NNFTs.

field theory can be thought of as a sloppy model in the sense discussed above since all of the momentum modes above some cutoff (set by the precision of our experimental apparatus) are in fact degenerate; they can be altered with impunity without impacting the efficacy of the model at experimentally accessible scales.

The above observation motivates the following interpretation; if we regard $\lambda_A^{-1}$ as the 'momentum' of the $A^{th}$ parameter then a stiff parameter corresponds to a low energy mode, while a sloppy parameter is a high energy mode. This identification becomes more than a mere analogy, however, when we make the following connection. Recall that the covariance of a local Euclidean field theory, viewed as a random process, is identified with the field propagator or Green function. That is, the covariance is precisely the operator inverse of the Laplacian. As we have established, the Fisher metric has an immediate interpretation as a covariance (49). Thus, the counterpart of the 'relevant Laplace operator' in the context of a generic statistical model is precisely the inverse of the Fisher metric; this justifies the interpretation of the reciprocal of the 'eigenvalues' $\lambda_A$ as generalized momenta.

### 3.3 Information shell renormalization for general inference problems and NN-FTs

Wilson's resolution to the problem of degeneracy in a continuum field theory was to perform an averaging or coarse graining over high energy modes. In its most straightforward form this corresponds to systematically integrating out degrees of freedom which depend on momenta above a sliding cutoff. In [30] a variation of this approach was applied to a NN using the insights described above. As a result, it was observed that the sloppy modes in the model were systematically removed. This 'information shell renormalization scheme' can be regarded as a perturbative form of the exact Bayesian renormalization scheme described in [31].[17] As the information shell scheme makes no assumptions about the underlying physical structure of the system under consideration it provides an ideal generalization of Wilson's renormalization procedure beyond the realm of local field theory. For this reason, the Bayesian inspired RG scheme is particularly well suited for initiating the study of renormalization for NNFTs.

We will now outline the information shell renormalization scheme to highlight its natural relation with NNFTs. Let $\pi_*(\theta)$ denote the posterior distribution after training, and denote by $\theta_* \equiv \mathrm{argmax}_{\theta \in M}(\pi_*(\theta))$ the maximum a posteriori parameter estimate (MAP).[18] Let $\mathcal{I}_{AB}^*$ denote the Fisher metric (50) evaluated at the MAP, and $\lambda_A^*$ denote its associated diagonal values. We can then specify a cutoff scale $\Lambda$ which splits the model space into a Cartesian product $M = M_\Lambda^< \times M_\Lambda^>$ where

$$M_\Lambda^< \equiv \{\theta^A \mid \lambda_A^* < \Lambda\}, \qquad M_\Lambda^> \equiv \{\theta^A \mid \lambda_A^* \geq \Lambda\}. \tag{53}$$

---

[17]The exact renormalization scheme is governed by a functional convection-diffusion equation on the posterior predictive distribution. In future work we plan to study the relationship between Bayesian renormalization and diffusion learning as facilitated by the NNFT correspondence.

[18]We should highlight that although we have stressed the perspective that the data $\pi_*(\theta)$ is obtained through a training procedure, we could just as well have provided $\pi_*(\theta)$ as initial data to perform the following renormalization procedure. For example, in a field theoretic context specifying $\pi_*(\theta)$ is tantamount to identifying the UV fixed point at the start of the RG flow.

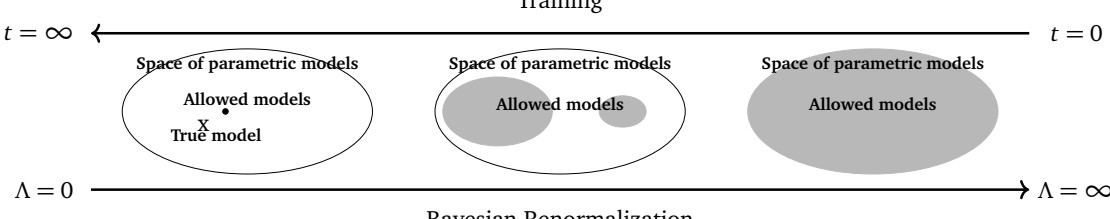

Figure 6: Schematic picture of information geometry interpretation of renormalization, where the top arrow represents the flow induced in model space by training, and the lower arrow the flow induced by Bayesian renormalization. We note that the space of allowed models may converge to a different point than the true model (X). In particular, for finite data, more than one epoch of training will place artificial importance on the sampled data. This may skew the final model away from the true model. However, in the limit of infinite data and a single training epoch, the final space of allowed models and the true model will coincide.

That is, $M_\Lambda^<$ consists of parameters that vary on 'length' scales smaller than $\Lambda$ as measured by $\mathcal{I}^*$. Conversely, $M_\Lambda^>$ consists of parameters that vary on scales greater than $\Lambda$. We then define the renormalized posterior distribution at scale $\Lambda$ as

$$\pi_\Lambda(\theta_<, \theta_>) = \pi_\Lambda^<(\theta_<)\pi_\Lambda^>(\theta_>), \quad \pi_\Lambda^<(\theta) \equiv \sqrt{\det \mathcal{I}_<^\Lambda(\theta_<)}, \quad \pi_\Lambda^> \equiv \int_{M_\Lambda^<} d^{n_<}\theta_< \, \pi_*(\theta_<, \theta_>). \quad (54)$$

Here, $\mathcal{I}_\Lambda^<(\theta_<)$ is the metric on $M_<^\Lambda$ induced by the full information metric on $M$ and $\pi_\Lambda^>$ is the posterior over $M_\Lambda^>$ obtained by marginalizing the full posterior over $M_\Lambda^<$. Thus, the renormalized posterior (54) is obtained by averaging over the sloppy parameters $M_\Lambda^<$ and subsequently assigning to these parameters the distribution of maximal ignorance – the Jeffreys prior – which is the information geometrically invariant form of a flat distribution.

The prescription (54) identifies $\Lambda$ as a proxy for the sensitivity of the model as measured by the relative entropy. Setting a cutoff $\Lambda$ corresponds to treating models whose relative entropy is less than $\Lambda$ as indistinguishable. For this reason, parameters whose characteristic radii fall below this scale present no relevant information at the desired level of accuracy/performance and thus should be coarse grained out of the model. Notice that in the limit as $\Lambda \to \infty$ the renormalized posterior converges to the full Jeffreys prior. This is consistent with the observation that at $\Lambda \to \infty$ all distributions are treated as effectively equivalent and thus the distribution over parameters should be the one that treats all models as equally likely (up to the problem of reparameterization invariance). A brief outline of this flow is provided in Figure 6.

The set of renormalized posterior distributions $\{\pi_\Lambda\}_{\Lambda \in \mathbb{R}}$ defines an RG flow. Each distribution moreover defines a renormalized posterior predictive distribution via (15):

$$p_\Lambda(y) \equiv \int_M d^n\theta \, \pi_\Lambda(\theta)p(y \mid \theta). \quad (55)$$

For a NN inference problem, the renormalized posteriors define renormalized $n$-point correlators

$$G_\Lambda^{(n)}(x_1, \ldots, x_n) = \mathbb{E}_{\pi_\Lambda}(\phi_\theta(x_1)\cdots\phi_\theta(x_n)). \quad (56)$$

Using the renormalized correlators (56) one can reconstruct the NNFT using the Edgeworth expansion (40) thereby resulting in a 'renormalized' NNFT, $p_\Lambda(\phi)$. The action of the theory at each $\Lambda$ is given explicitly by $S_\Lambda(\phi) = S_\Lambda^{\text{free}}(\phi) + S_\Lambda^{\text{int}}$ with

$$S_\Lambda^{\text{free}}(\phi) = \int d^d x_1 \, d^d x_2 \, \phi(x_1) G_{\Lambda,c}^{(2)}(x_1, x_2)^{-1} \phi(x_2), \tag{57}$$

$$S_\Lambda^{\text{int}}(\phi) = \sum_{n=3}^{\infty} \int \prod_{j=1}^{n} d^d x_i \, g_\Lambda^{(n)}(x_1, \ldots, x_n) \phi(x_1) \cdots \phi(x_n), \tag{58}$$

where the renormalized couplings $g_\Lambda^{(n)}$ are obtained from (56) via the inverse Feynman rules.

We should emphasize that the one parameter family $S_\Lambda$ describes a flow in the space of NNFTs induced by information theoretic coarse graining. We refer to this as a (Bayesian) RG flow, and, by the arguments of this section, claim that it is a suitable replacement for a standard field theoretic RG flow based on notions of spatial locality when the conventional techniques originally tailored to physical field theories are not applicable.

## 4 Demonstrations

This section is split into two parts; in Section 4.1, we provide an *analytic exploration* of Bayesian renormalization for infinite width NNs of arbitrary depth. In Section 4.2 we present a *numerical exploration* of the very same setup, with special attention paid to important practical considerations. Although the two parts of this section are closely related, we would like to stress that they can be read independently. Thus, those who are interested only in the analytic exercise may read only Sections 4.1 and 4.1.1 and move on; likewise, those who are interested only in the numerical exercise may read only Section 4.2.

Focusing first on Section 4.1, it has been shown that, under suitable conditions, infinite width networks of arbitrary depth can be described as Gaussian Random Processes [72, 76, 98–105]. Provided training data is Gaussian, Bayesian conjugacy implies that this correspondence holds not only at initialization, but also for all stages throughout training [102]. In this respect, the renormalization of such a network coincides conceptually with the renormalization of a free field theory. The analysis of Section 4.1 is valid for infinite-width networks with arbitrary activation functions. In previous work it has been shown that an infinite width network with cosine activation functions is explicitly dual to a massive free scalar (Euclidean) field theory [32, 33]. Specializing to this case, in Section 4.1.1 we demonstrate that BRG applied to such an SFT results in a flow of the mass, similarly to the standard renormalization of a massive free field.

Moving to Section 4.2, we study the training and subsequent Bayesian renormalization of an ensemble of asymptotically wide NNs. The use of an ensemble (in lieu of analytic expressions for parameter and predictive distributions) and asymptotic networks (in lieu of infinite width networks) provide computationally flexible avenues for implementing the procedures described in previous sections in the presence of real data and feasible model architectures. Using the NNFT correspondence, we verify that the posterior predictive distribution is Gaussian to leading order, and construct information theoretically renormalized parameter distributions as a function of a flowing distinguishability scale. As expected, the performance of the network is relatively unchanged under this renormalization up to a critical scale during which time only sloppy parameters are being coarse grained out of the model. Moving beyond this scale, the network performance is significantly impacted as stiff parameters start to be removed.

## 4.1  Bayesian RG for NNGPs

In this section, we present a simple instance of information theoretic RG flow in an infinite width i.i.d. initialized NN, dual to a generic free field theory. This discussion brings forth operator renormalization of inverse propagators in free field theories, through information geometric coarse graining of dual infinite width i.i.d. initialized NNs (with GRP priors), trained under the strict constraints given in [103]. The Bayesian framework guarantees that the posterior predictive distribution, associated with an exact GRP prior and normally distributed data, e.g. having a Gaussian likelihood, will remain in the class of GRPs at every stage throughout training [102]. Translated into the NN training point of view, a NN with GRP prior trained via stochastic gradient descent with $L2$ loss will remain in the class of GRPs throughout training, and thus is characterized by a changing mean function and covariance. Strictly speaking, such networks require infinite widths and are therefore not realizable practically; nonetheless, asymptotically large widths in hidden layers provides a good approximation [103]. We should note, however, that if these assumptions are violated, e.g. if the likelihood is non-Gaussian or the loss is non-quadratic, then the NN distributions will acquire non-trivial higher cumulants over the course of training.

Let us start with a feed forward NN of arbitrary depth $L$. The width of the $\ell^{th}$ layer, which we denote by $O_\ell$, is given by $n_\ell$. In other words, one may regard $O_\ell \sim \mathbb{R}^{n_\ell}$. The first layer is taken to be the input space, $O_0 = I$, and the final layer the output space, $O_L = O$. The network architecture is defined recursively by[19]

$$
\begin{aligned}
\phi_{(w,b)}^{(1)}(x) &\equiv b_1 + w_1(x), \\
h_{(w,b)}^{(1)}(x) &\equiv \sigma\left( \phi_{(w,b)}^{(1)}(x) \right), \\
\phi_{(w,b)}^{(\ell)}(x) &\equiv b_\ell + w_\ell\left( h_{(w,b)}^{(\ell-1)}(x) \right), \\
h_{(w,b)}^{(\ell)}(x) &\equiv \sigma\left( \phi_{(w,b)}^{(\ell)}(x) \right),
\end{aligned}
\tag{59}
$$

with final output

$$
\phi_{(w,b)}(x) \equiv \phi_{(w,b)}^{(L)}(x).
\tag{60}
$$

Here, $b_\ell \in O_\ell$ (i.e. a $n_\ell$-vector), $w_\ell : O_{\ell-1} \to O_\ell$ is a linear map (i.e. an $n_\ell \times n_{\ell-1}$ matrix), and $\sigma$ is an arbitrary element-wise non-linearity.

At initialization, we take our network parameters to be normally distributed as[20]

$$
b_\ell^{i_\ell} \sim \mathcal{N}(0, \sigma_{b_\ell}^2), \ \forall i_\ell, \qquad w_{\ell\ i_{\ell-1}}^{i_\ell} \sim \mathcal{N}(0, \sigma_{w_\ell}^2), \ \forall i_\ell, i_{\ell-1}.
\tag{61}
$$

Here we have used the notation $y \sim \mathcal{N}(\mu, \sigma^2)$ to mean that $y$ is a random variable sampled from a normal distribution with mean $\mu$ and variance $\sigma^2$. In words, (61) means that the parameters in each layer are each sampled from the same distribution (identically distributed), and that each layer is independent from every other layer. The final output of the network is given by

$$
\phi_{(w,b)}^{i_L}(x) = b_L^{i_L} + \sum_{i_{L-1}=1}^{n_{L-1}} w_{L\ i_{L-1}}^{i_L} h_{(w,b)}^{(L-1)}(x)^{i_{L-1}}.
\tag{62}
$$

Under the choice (61) the second term in (62) is a sum of independent and identically distributed random variables. Thus, in the limit that $n_L$ is sufficiently large the distribution of

---

[19]We also take $\phi_{(w,b)}^{(0)}(x) = x$.

[20]We have introduced notation in which $i_\ell = 1,\ldots,n_\ell$ is an index for the space $O_\ell$ in a chosen basis. Thus, $b_\ell^{i_\ell}$ can be interpreted as the $i_\ell^{\text{th}}$ bias parameter in the $\ell^{\text{th}}$ layer.

this term will be Gaussian. The first term in (62) is also a normal random variable, and thus the output of the network – as the sum of two normal random variables, will itself be normal (conditional on each input $x$). As a result, the distribution over $\phi(x)$ is a Gaussian random process (GRP) with kernel determined via (61).

Bayesian inference on infinitely wide NNs is well studied, and it is known that the posterior predictive distribution remains Gaussian in the strict infinite width limit [102–105]. Remarkably, if one freezes all but the parameters of the final layer $L$ after initialization, the training of the network via gradient descent coincides with the training dynamics of the network's linearized counterpart [102].[21] What's more, in the infinite width limit the training of this linearized model reproduces the posterior predictive distribution in probability in the limit of large training times [102]. In this section we will use these observations to inspire a Bayesian analysis of the GRP as follows – we freeze the distribution over the hidden layers $\ell < L$ at their initialization distribution and regard the inference procedure as updating only the distribution over the parameters $(b_L, w_L)$. We denote by $\pi^{(L)}(b_L, w_L)$ the posterior distribution over the parameters of the output layer. We will also denote by $\pi^{(-L)}(b^{(-L)}, w^{(-L)})$ the distribution over the remaining parameters which will be as defined in (61). The full posterior is therefore of the form of a product

$$\pi(b, w) = \pi^{(L)}(b_L, w_L)\pi^{(-L)}(b^{(-L)}, w^{(-L)}), \tag{63}$$

indicating that the final layer and the intermediate layers remain independent over the course of the training.

In terms of the posterior (63) we can now compute directly the parameters of the posterior predictive GRP. The mean function is given by

$$\begin{aligned}
\mu_\pi^{i_L}(x) &\equiv \mathbb{E}_\pi\left(\phi_{(w,b)}^{i_L}(x)\right) \\
&= \mathbb{E}_{\pi^{(L)}}(b_L^{i_L}) + \mathbb{E}_{\pi^{(L)}}(w_{L\,i_{L-1}}^{i_L})V^{i_{L-1}}(x).
\end{aligned} \tag{64}$$

The covariance is given by

$$\begin{aligned}
\Sigma_\pi^{i_L j_L}(x, y) &\equiv \mathrm{Cov}_\pi\left(\phi_{(w,b)}^{i_L}(x), \phi_{(w,b)}^{j_L}(y)\right) \\
&= \mathrm{Cov}_{\pi^{(L)}}(b_L^{i_L}, b_L^{j_L}) \\
&\quad + \mathrm{Cov}_{\pi^{(L)}}(b_L^{i_L}, w_{L\,j_{L-1}}^{j_L})V^{j_{L-1}}(y) + \mathrm{Cov}_{\pi^{(L)}}(w_{L\,i_{L-1}}^{i_L}, b_L^{j_L})V^{i_{L-1}}(x) \\
&\quad + \mathrm{Cov}_{\pi^{(L)}}(w_{L\,i_{L-1}}^{i_L}, w_{L\,j_{L-1}}^{j_L})V^{i_{L-1}j_{L-1}}(x, y) \\
&\quad + \mathbb{E}_{\pi^{(L)}}(w_{L\,i_{L-1}}^{i_L})\mathbb{E}_{\pi^{(L)}}(w_{L\,j_{L-1}}^{j_L})\left(V^{i_{L-1}j_{L-1}}(x, y) - V^{i_{L-1}}(x)V^{j_{L-1}}(y)\right).
\end{aligned} \tag{65}$$

In (64) and (65) we have defined (conditional on inputs $x, y \in I$)

$$V^{i_{L-1}}(x) \equiv \mathbb{E}_{\pi^{(-L)}}\left(h_{(w,b)}^{(L-1)}(x)^{i_{L-1}}\right), \tag{66}$$

$$V^{i_{L-1}j_{L-1}}(x, y) \equiv \mathbb{E}_{\pi^{(-L)}}\left(h_{(w,b)}^{(L-1)}(x)^{i_{L-1}}h_{(w,b)}^{(L-1)}(y)^{j_{L-1}}\right). \tag{67}$$

Given posterior $\pi$ the NN function $\phi$ is therefore governed by a posterior predictive GRP:

$$\phi \sim \mathrm{GRP}(\mu_\pi, \Sigma_\pi). \tag{68}$$

---

[21]This result is proven for mean squared losses and in a particular learning regime dictated by a critical learning rate. See [102] for a more detailed discussion.

To proceed we make a simplifying assumption that the terminal posterior $\pi^{(L)}(b_L, w_L)$ is a Gaussian sharply peaked around the optimal gradient descent estimate $\theta_* \equiv (\overline{b}_L, \overline{w}_L)$.[22] That is, of the form

$$\pi^{(L)}(b_L, w_L) \sim \mathcal{N}(\theta_*, \Xi), \tag{69}$$

in which the covariance is split as

$$\Xi = \begin{pmatrix} \Xi_{bb} & \Xi_{bw} \\ \Xi_{wb} & \Xi_{ww} \end{pmatrix}. \tag{70}$$

The Ansatz (69) is consistent with large deviations theory which dictates that the distribution over fitted parameters should become Gaussian in the limit that the amount of collected data becomes large. In fact, in this limit the covariance (70) will be directly related to the Fisher information metric of the model class (68).[23] Given (69) we can rewrite (64) and (65) as

$$\mu_\pi^{i_L}(x) = \overline{b}_L^{i_L} + \overline{w}_{L\ i_{L-1}}^{i_L} V^{i_{L-1}}(x),$$

$$\Sigma_\pi^{i_L j_L}(x, y) = \Xi_{bb}^{i_L j_L} + \Xi_{bw\ j_{L-1}}^{i_L j_L} V^{j_{L-1}}(y) + \Xi_{wb\ i_{L-1}}^{i_L\ j_L} V^{i_{L-1}}(x)$$

$$+ \Xi_{ww\ i_{L-1}}^{i_L\ \ j_L}{}_{j_{L-1}} V^{i_{L-1}j_{L-1}}(x, y) \tag{72}$$

$$+ \overline{w}_{L\ i_{L-1}}^{i_L} \overline{w}_{L\ j_{L-1}}^{j_L} \left( V^{i_{L-1}j_{L-1}}(x, y) - V^{i_{L-1}}(x) V^{j_{L-1}}(y) \right).$$

To simplify our notation and facilitate a more interpretable analysis we will hereafter work in the case where the biases $b_L = 0$. If one would like, they can think of the biases as being absorbed in the weights with an appropriate elongation $h(x) \mapsto (h(x), 1)$. In this case we will no longer need to worry about the block structure of (70) which is reduced to only $\Xi_{ww}$ which we will now denote simply by $\Xi$. Thus, (72) reduces to

$$\mu_\pi^{i_L}(x) = \overline{w}_{L\ i_{L-1}}^{i_L} V^{i_{L-1}}(x),$$

$$\Sigma_\pi^{i_L j_L}(x, y) = \Xi^{i_L\ \ j_L}{}_{i_{L-1}\ j_{L-1}} V^{i_{L-1}j_{L-1}}(x, y) \tag{73}$$

$$+ \overline{w}_{L\ i_{L-1}}^{i_L} \overline{w}_{L\ j_{L-1}}^{j_L} \left( V^{i_{L-1}j_{L-1}}(x, y) - V^{i_{L-1}}(x) V^{j_{L-1}}(y) \right).$$

In this context a Bayesian renormalization scheme can be described purely at the level of the hyperparameters with the effect of this renormalization on the posterior predictive distribution exactly computable via (73).

In particular, let $\mathcal{I}_{i_L}{}^{i_{L-1}}{}_{j_L}{}^{j_{L-1}}$ be the Fisher information metric evaluated at the gradient descent estimator $\theta_*$. Here we are using the components of the weight matrix, $w_{L\ i_{L-1}}^{i_L}$ as coordinates for the information geometry associated with the family of parameterized models for the network $\phi_w$. Let $\lambda_{i_L}{}^{i_{L-1}} \equiv \mathcal{I}_{i_L}{}^{i_{L-1}}{}_{i_L}{}^{i_{L-1}}$. At scale $\Lambda$ we take the following assignment for the hyperparameters of the renormalized posterior:

$$\overline{w}_L(\Lambda)^{i_L}{}_{i_{L-1}} \equiv \begin{cases} \overline{w}_{L\ i_{L-1}}^{i_L}, & \lambda_{i_L}{}^{i_{L-1}} \geq \Lambda, \\ 0, & \lambda_{i_L}{}^{i_{L-1}} < \Lambda, \end{cases} \tag{74}$$

$$\Xi(\Lambda)^{i_L}{}_{i_{L-1}}{}^{j_L}{}_{j_{L-1}} \equiv \begin{cases} \Xi^{i_L}{}_{i_{L-1}}{}^{j_L}{}_{j_{L-1}}, & \lambda_{i_L}{}^{i_{L-1}} \geq \Lambda, \text{ and } \lambda_{i_L}{}^{i_{L-1}} \geq \Lambda, \\ \delta^{i_L}{}_{i_{L-1}}{}^{j_L}{}_{j_{L-1}} \sigma^2_{w_L}, & \lambda_{i_L}{}^{i_{L-1}} < \Lambda, \text{ or } \lambda_{j_L}{}^{j_{L-1}} < \Lambda. \end{cases} \tag{75}$$

---

[22]We note that here $\overline{b}_L, \overline{w}_L$ are the hyperparameters which identify the mean of the parameter distributions for biases and weights. In practical applications, $\overline{b}_L, \overline{w}_L$ can be estimated via ensemble averaging.

[23]This result goes under the name of the Bernstein-von Mises theorem. It states that, under sufficient regularity conditions, the posterior distribution relative to a collection of data $D = \{y_1, \ldots, y_{|D|}\}$ will be of the form

$$\pi_D(\theta) = \mathcal{N}(\theta \mid \theta_*, (|D|\mathcal{I}_*)^{-1}), \tag{71}$$

where $\theta_*$ is the parameter associated with the data generating distribution and $\mathcal{I}_*$ is the Fisher metric on model space evaluated at $\theta_*$.

In words, for weights which are sloppy at the scale $\Lambda$, the mean value is set equal to zero (on average the weight is 'turned off'), the weight is decorrelated from the rest of the parameters, and its covariance is set equal to a constant value $\sigma_{w_L}^2$ which is the same for all sloppy weights. Notice that in the limit as $\Lambda \to \infty$ the distribution over parameters reverts back to the form of the prior in which all weights are regarded as independent and identically distributed normal random variables.

The scheme described in (74) and (75) differs slightly from the more general one described by equation (54). Rather than reverting the distribution over sloppy parameters back to the Jeffreys prior, we have reverted these paramters back to a Gaussian prior. This is more natural for inference over NNs in which the Jeffreys prior is generally quite untenable. Nevertheless, provided the variance parameter $\sigma_{w_L}^2$ is taken to be large the resulting distribution over sloppy parameters will be roughly 'flat'. In any case, given (74) and (75) we now have a closed form for the renormalized posterior predictive distribution at each scale $\Lambda$: $\phi_\Lambda \sim \mathrm{GRP}(\mu_{\pi_\Lambda}, \Sigma_{\pi_\Lambda})$ with

$$\mu_{\pi_\Lambda}^{i_L}(x) = \overline{w}_L(\Lambda)^{i_L}{}_{i_{L-1}} V^{i_{L-1}}(x),$$

$$\Sigma_{\pi_\Lambda}^{i_L j_L}(x, y) = \Xi(\Lambda)^{i_L}{}_{i_{L-1}}{}^{j_L}{}_{j_{L-1}} V^{i_{L-1} j_{L-1}}(x, y)$$
$$+ \overline{w}_L(\Lambda)^{i_L}{}_{i_{L-1}} \overline{w}_L(\Lambda)^{j_L}{}_{j_{L-1}} \left( V^{i_{L-1} j_{L-1}}(x, y) - V^{i_{L-1}}(x) V^{j_{L-1}}(y) \right). \tag{76}$$

### 4.1.1 BRG as mass renormalization of a generalized cos-net architecture

To better understand Bayesian RG flows (76) of NN GRPs let us now provide a concrete worked example. Here, information geometric RG over an i.i.d. initialized infinite width NN amounts to the mass renormalization in dual action in the conventional field theoretic sense. To that end, we extend the studies of Section 4.1 to the case of a single hidden layer architecture at infinite width $\lim N \to \infty$, one-dimensional output, and a generalized cos-net activation from [33], having covariance spectrum identical to that of free scalar field theory on Euclidean space. The network output

$$\phi_{w,b,c}(x) = w_i h^i, \qquad h^i(x) = \sum_j \frac{\cos(b^i{}_j x^j + c^i)}{\sqrt{\sum_k (b^i{}_k)^2 + m^2}}, \tag{77}$$

is based on $w$, the final (trained) layer weights, and $(b, c)$, the weights and biases of the hidden layer, respectively. At initialization $\pi(w, b, c) = \pi_G(w)\pi_G(b)\pi_G(c)$, with

$$\pi_G(w) = \prod_i e^{-\frac{N}{2\sigma_w^2} w_i w_i}, \tag{78}$$

$$\pi_G(b) = \prod_i \pi_G(\mathbf{b}_i), \quad \text{with} \quad \pi_G(\mathbf{b}_i) = \mathrm{Unif}(B_R^d), \tag{79}$$

$$\pi_G(c) = \prod_i \pi_G(c_i), \quad \text{with} \quad \pi_G(c_i) = \mathrm{Unif}([-\pi, \pi]). \tag{80}$$

In [33] it has been shown that, with standard initialization distribution, the generalized cos-net leads to a GRP whose covariance has Fourier transform

$$\Sigma(p) = \frac{C_d \, \sigma_w^2}{p^2 + m^2}. \tag{81}$$

In (81), $\sigma_w^2$ is the covariance of the Gaussian distribution for each weight $w$ and $C_d = \frac{(2\pi)^d}{2 \operatorname{Vol}(B_R^d)}$ is a constant with $B_R^d$ the $d$-dimensional ball of radius $R$. Here, $m, R$ are network hyperparmeters playing the roles of mass and ultraviolet cutoffs, respectively.

Rescaling the field as

$$\phi_{w,b,c} \mapsto \tilde{\phi}_{w,b,c} = \frac{1}{\sqrt{C_d \, \sigma_w^2}} \phi_{w,b,c} \, , \tag{82}$$

(81) can be recast in the form

$$\tilde{\Sigma}(p) = \frac{1}{p^2 + m^2} \, . \tag{83}$$

A GRP with kernel (83) has a density $p(\tilde{\phi}) \sim \exp(-S[\tilde{\phi}])$, where

$$S[\tilde{\phi}] = \int d^d x \, \tilde{\phi}(x)(\nabla^2 + m^2)\tilde{\phi}(x) , \tag{84}$$

is the action of a free scalar field theory with mass $m$.

Now, let $\pi_\Lambda(w, b, c) = \pi_\Lambda^{(L)}(w)\pi_{init}^{(-L)}(b, c)$ be the probability distribution over parameters at Bayesian renormalization scale $\Lambda$. Per our above analysis the predictive distribution associated with this parameter distribution is given by (76), where $\overline{w}(\Lambda)$ and $\Xi(\Lambda)$ are determined via training along with the Bayesian RG scheme dictated in (74) and (75). Thus, it remains only to compute the functions $V^i(x)$ and $V^{ij}(x, y)$ which are defined entirely by the activation and the fixed distribution $\pi_{init}(b, c)$. For the initialization distribution quoted in [33] we can compute these directly as

$$V^i(x) = 0 , \qquad V^{ij}(p) = \delta^{ij} \frac{C_d}{p^2 + m^2} \, . \tag{85}$$

Here we have written the Fourier transform of the two point function.

Plugging (85) back into (76) we obtain a one parameter family of generalized cos-net predictive distributions of the form $p(\tilde{\phi}) \sim \mathrm{GRP}(\mu_{\pi_\Lambda}, \Sigma_{\pi_\Lambda})$ with

$$\mu_{\pi_\Lambda}(x) = 0 , \qquad \Sigma_{\pi_\Lambda}(p) = \frac{C_d \, \sigma_w(\Lambda)^2}{p^2 + m^2} \, , \tag{86}$$

where we have defined

$$\sigma_w(\Lambda)^2 \equiv \sum_i \Xi(\Lambda)_{ii} + \overline{w}(\Lambda)_i^2 \, . \tag{87}$$

Notice that (86) is of the same general form as (81) with a new *effective variance* $\sigma_w^2(\Lambda)$ for last layer weights. The resulting GRPs may therefore be interpreted as free field theories for a field $\phi_\Lambda$ which is rescaled at each Bayesian RG scale as:

$$\phi_\Lambda = \frac{1}{\sqrt{C_d \, \sigma_w^2(\Lambda)}} \phi \, , \tag{88}$$

or equivalently as a free field theory in which the momentum and mass have been renormalized as

$$p^2 \mapsto p_\Lambda^2 = \frac{p^2}{C_d \, \sigma_w^2(\Lambda)} \, , \qquad m^2 \mapsto m_\Lambda^2 = \frac{m^2}{C_d \, \sigma_w^2(\Lambda)} \, . \tag{89}$$

More succinctly, the Bayesian renormalization of the generalized cos-net architecture can be absorbed into a redefinition of the hyperparameter $R$ which sets a UV scale for the theory in the traditional sense as a momentum cutoff. In particular,

$$C_d \mapsto C_d(\Lambda) = \frac{\sigma_w(\Lambda)^2 (2\pi)^d}{2\mathrm{Vol}(B_R^d)} = \frac{(2\pi)^d}{\mathrm{Vol}(B_{R_\Lambda}^d)} \, . \tag{90}$$

In words, the volume of modes in momentum space which are allowed to contribute to the theory is decreased by a factor of $\sigma_w(\Lambda)^2$. If $\sigma_w(\Lambda)^2$ increases under the RG flow this will have precisely the effect of shrinking the set of modes which contribute, thereby inducing a genuine Wilsonian momentum shell RG scheme.

## 4.2 Experimental implementation

While prior discussions in this work have been largely abstract and theoretical in nature, this section presents an experimental implementation of Bayesian renormalization in the context of an NNFT. In Section 4.1, we considered networks in the infinite width limit; for obvious reasons, implementing this practically is not possible. Instead, one considers 'asymptotically wide' networks with very large[24] final layers ($N = 2000$).

In this section, we consider the training and Bayesian renormalization of an ensemble of asymptotically wide NNs. In principle, as in the previous sections, we would prefer to work directly in terms of a predictive distribution over networks. However, as it is computationally infeasible to perform exact Bayesian inference in this context, we pass to an ensemble of networks whose statistics serve as a proxy for the aforementioned distribution. In fact, this is one of the advantages of the NNFT correspondence as the statistics of our ensemble (e.g. its numerical moments and cumulants) can be directly translated into an (approximate) analytic form for the predictive distribution via (41). Thus, in our numerical analysis, one goal will be to demonstrate how this is done in a simple case in which the network ensemble is assumed to be a GRP at all stages of training.

In particular, we begin with an ensemble of asymptotically wide NNs, each with 2000 neurons in the final layer. It is known that in the infinite width limit, when the network is trained subject to an $L^2$ loss and equipped with sufficiently well-behaved activations, the posterior predictive distribution of the network outputs is guaranteed to be a Gaussian random process (GRP). Thus, the posterior predictive distribution of an asymptotically wide network will also correspond to a GRP at leading order. In the following experiments, we use NNFT to approximate the posterior predictive distribution (assuming the network is sufficiently large to observe the aforementioned GRP behaviour) using the first two cumulants of the distribution of ensemble evaluations. We then perform the Bayesian renormalization procedure described in Section 4.1, and track the resulting flow in the distribution.

### 4.2.1 Implementation and network architecture

As in Section 4.1, we consider a recursively defined fully connected feed-forward multi-layer perceptron NN with non-trivial activations internally, and identity activations on the layers adjacent to the input and output.[25] Specializing to the network architecture proposed in [102], we consider a three layer network (i.e. $L = 3$), with internal dense layer widths $(n_1, n_2, n_3) = (10, 10, 2000)$. In accordance with [102], the final layer is suitably sized to facilitate large width effects, and thus we expect the predictive distribution over the network should be a GRP both at initialization and throughout training.

We expect that our results should generalize to arbitrary learning problems, and well optimized training regimes as none of our conclusions depend on these choices. For these reasons the particular choice of problem and specifics of the training/performance of the model are not significant in our analysis. The purpose of this section is to emphasize the interpretability of our approach to training and renormalization rather than to compete with modern state-of-the-art NNs. For our sample problem we consider learning the 2D quadratic,

$$\phi^*(x_1, x_2) := (a - x_1)^2 + b(x_2 - x_1^2). \tag{91}$$

---

[24]In general, the meaning of 'large' is highly architecture and problem dependent, although as a rule of thumb a final layer size of 2000 or greater is typically considered sufficient [102] to see large width effects.

[25]It is typical for networks equipped with ReLU activations to have identity functions on the input and output in order to facilitate negative values.

Table 3: Table of model hyperparameters.

| Network parameter | Value |
|---|---|
| Optimiser | Gradient Descent |
| Epochs | 1000 |
| Learning rate | $10^{-6}$ |
| Training samples | 200 |
| Validation samples | 200 |
| Internal Layer 1 width | 10 |
| Internal Layer 2 width | 10 |
| Internal Layer 3 width | 2048 |
| Internal activation function | ReLU |

The input in this case is two-dimensional, $x \in I = \mathbb{R}^2$, and is sampled from the Gaussian distribution $X \sim \mathcal{N}(0, \mathbb{I}_2)$, where $\mathbb{I}_2$ is the standard $2 \times 2$ identity matrix. Training is performed such that the network output $\phi_\theta(x)$, where $\phi_\theta$ is the network function parameterized by $\theta$, minimizes the $L^2$ loss function,

$$\mathcal{L}[\theta](x, y) := |y - \phi_\theta(x_1, x_2)|^2 , \tag{92}$$

for the desired target, $y \in \mathbb{R}$, corresponding to the input, $x$. Optimization is performed using 'vanilla' gradient descent.[26]

It is typical in the training of NNs to start with a collection of random draws from a seed distribution over parameters, performing the chosen optimization procedure for each parameter draw in parallel. For our purposes, as was discussed in Section 2, it is natural to interpret the seed distribution as a prior over networks, and to treat the ensemble of parallel networks as realizations from the Bayesian posterior at each stage of training. In this way, we can approximate the posterior in a numerically tractable form by parsing calculable statistics from the ensemble. For our sample problem, we take the prior distribution to be Gaussian. Following the notation of Section 2, we denote the ensemble of parameters by $\{\theta_n(D)\}_{n=1}^k$, where $D = \{x_i, \phi_i^*\}_{i \in \mathcal{I}}$ is a training sample. In this case, the training is supervised, so each network input datum, $x_i$, is matched with the corresponding ground truth model realization, $\phi_i \equiv \phi^*(x_i)$.

As we have mentioned, to good approximation we expect the posterior predictive distribution over network outputs to be Gaussian both at initialization and after training. Thus, in this case we should be able to recover the full distribution by computing the arithmetic mean and covariance over outputs relative to the realized parameter ensemble $\{\theta_n(D)\}_{n=1}^k$. This is a particularly simple application of the NNFT correspondence in which the first two cumulants are sufficient for performing the Edgeworth expansion.[27] In Section 4.1.1 we highlighted the analysis of the generalized cos-net for the striking similarity of its associated GRP to that of a free scalar field theory. However, while prior works have shown promising results from networks with cosine activations at initialization, they generally train poorly (due to their periodic nature). Therefore, we only consider NNs with ReLU activations in this numerics section. However, we highlight that the analysis of Section 4.1 is sufficiently general as to apply to any activation provided one can compute the functions (66).

---

[26]By 'vanilla' gradient descent, we mean traditional gradient descent as opposed to stochastic gradient descent.

[27]In fact, practically speaking, expanding probability distributions in terms of their $n$-point functions generalises well to many realistic datasets, even for sensibly low $n$ [106].

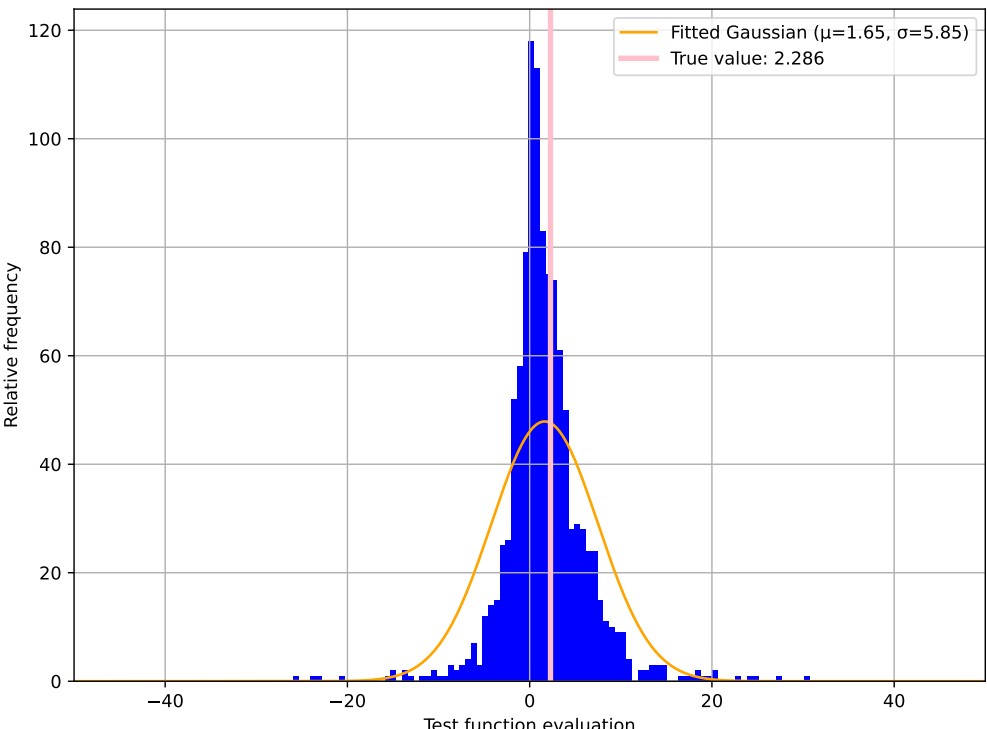

Figure 7: Distribution of evaluations of the test function at a characteristic valida-
tion sample fitted with a Gaussian distribution prior to renormalization, with mean
approximately equal to the analytic value.

### 4.2.2 Training

To mitigate confusion, we emphasize that training and renormalization of the network are
disparate processes – in the current form, the network is first trained conventionally, then
renormalized (in 4.2.3). While it is indeed true that the Bayesian renormalization scheme
may be applied at any stage during the training, even at initialization, in practice relevant
(and, inversely, irrelevant) parameters are typically not distinguished until near the end of
training when the loss has stabilized.[28]

For concreteness, we consider training the network to learn the function (91) with
$(a, b) = (1, -0.5)$. Figure 7 shows the empirical distribution over outputs for a fixed input. In
line with our expectations, this distribution is well approximated by a Gaussian peaked around
the value of the target function evaluated on this input.

### 4.2.3 Renormalization

Let $\phi_j(x) := \phi_{\theta_j}(x)$ be the output of the network evaluated on the $j^{th}$ parameter estimate of
the ensemble for some fixed input $x \in I$.[29] In this notation we obtain an ensemble of network
realizations as

$$\Phi(x) := \{\phi_{\theta_j}(x) \mid \theta_j \in \{\theta_n\}_{n=1}^k\}, \tag{93}$$

for each input, $x$.

---

[28]This phenomena is particularly manifest in 'grokked' networks [107,108]: models undergoing grokking rapidly
change between memorization and generalization phases during training. In this case, the parameter relevance is
not stable until post-grokking.

[29]In this section we will abbreviate $\theta_j(D)$ to $\theta_j$ for notational simplicity.

For each member of the ensemble we can now perform a Bayesian renormalization as described at the beginning of this section. This renormalization proceeds in a series of three steps:

1. For each $\theta_j$ we compute an associated Fisher metric with diagonal elements $\lambda^i_{(j)}$ for each parameter direction $i$.

2. Fixing a cutoff $\Lambda$, we partition each parameter tuple $\theta_j$ as

$$\theta_j = (\theta_j^<(\Lambda), \theta_j^>(\Lambda)), \tag{94}$$

with

$$\theta_j^<(\Lambda) \equiv \{\theta_j^i \mid \lambda^i_{(j)} < \Lambda\}, \qquad \theta_j^>(\Lambda) \equiv \{\theta_j^i \mid \lambda^i_{(j)} \geq \Lambda\}. \tag{95}$$

3. Following (54), we define the renormalized parameter $\tilde{\theta}_j(\Lambda)$ by retaining the 'relevant' parameter directions $\theta_j^>(\Lambda)$ and resampling the 'irrelevant' parameter directions from their prior distribution[30]

$$\tilde{\theta}(\Lambda) \equiv (\tilde{\theta}_j^<(\Lambda), \theta_j^>(\Lambda)), \qquad \tilde{\theta}_j^<(\Lambda) \sim \pi_0. \tag{96}$$

At each scale $\Lambda$ we obtain an ensemble of renormalized networks:

$$\Phi_\Lambda(x) = \{\phi_{\tilde{\theta}_j(\Lambda)}(x)\}_{j=1}^k. \tag{97}$$

To demonstrate experimentally the notion of 'sloppiness', it is instructive to consider (96) for a set of cutoffs $\Gamma = \{0, \Lambda_1, \ldots, \Lambda_L\}$: this results in a collection of model ensembles for each cutoff,

$$\Phi_\Gamma = \{\Phi, \Phi_{\Lambda_1}, \ldots, \Phi_{\Lambda_L}\}, \tag{98}$$

where the first element is the fully trained ensemble prior to renormalization.

Figure 8 depicts the model loss averaged over all networks in the ensemble for a set of cutoffs $\Lambda \in \Gamma_{\text{Exp}}$. Notice that up to the 'critical scale' $\Lambda_C$ the model loss is practically unchanged despite the fact that the training has been reverted for a large number of parameter directions. This corroborates the claim that the parameters being coarse grained over up to this scale are largely 'irrelevant' modes for which the parameter values suggested by training are not well constrained. To the right of this critical scale, however, the loss starkly increases as the renormalization procedure begins to remove relevant parameters.

One may also track the distribution of realizations of the network ensemble as the models undergo Bayesian renormalization. Figure 9 shows a histogram of the empirical distribution over outputs on either side of the critical cutoff. As expected from the analysis in Section 4.1, the mean and variance of the distribution are renormalized over the duration of the flow – this phenomena can be seen more clearly in Figure 10. We note a large increase in the variance of the distribution after the critical cutoff, which is consistent with Figure 8. This observation is also depicted in Figure 10a.

## 5 Discussion

The story of this paper is told through the commutative diagram seen in Figures 1 and 2, repeated below for convenience.

---

[30]This approach differs slightly from that described in (54) in which the distribution over irrelevant parameters is reverted to the Jeffrey's prior. In this case it is more conceptually and computationally straightforward to resample from the chosen prior.

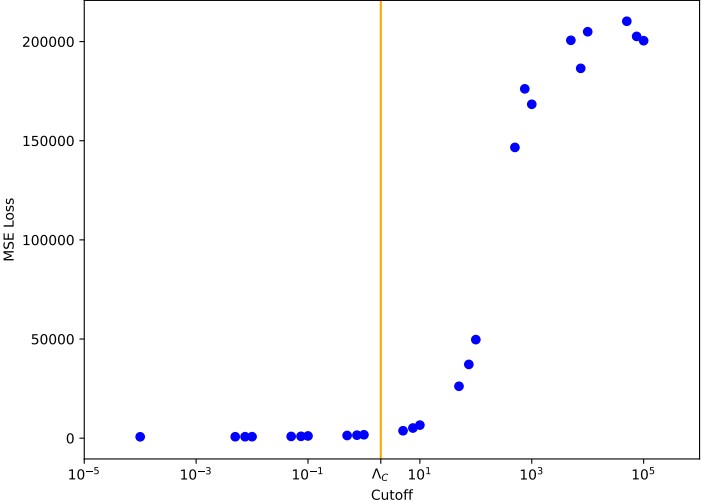

Figure 8: Average model loss as a function of cutoff (note the logarithmic scale on the horizontal axis).

$$
\begin{array}{ccc}
(\phi_\theta, \pi) & \xrightarrow{NNFT} & S[\phi] \\
\downarrow{\scriptstyle BRG} & & \downarrow{\scriptstyle BRG} \\
(\phi_\theta, \pi_\Lambda) & \xrightarrow{NNFT} & S_\Lambda[\phi]
\end{array}
$$

To briefly review, the main legs of the diagram coincide with BRG and NNFT. As reviewed in Section 3.1, NNFT maps pairs $(\phi_\theta, \pi)$, comprising a NN architecture and its parameter distribution, into the space of SFTs. More explicitly, NNFT constructs an SFT with action, $S[\phi]$, whose correlation functions are equivalent to those obtained from an infinite ensemble of NN output functions. BRG (as reviewed in Section 3.2) is a map from a given parameter distribution, $\pi$, into a coarse-grained version, $\pi_\Lambda$. Here, coarse graining is defined as averaging over fine-grained information in parameter space as determined by the Fisher information metric. The Fisher information metric measures the level of distinguishability between models and the cutoff $\Lambda$ sets the desired scale of distinguishability. Thus, $\pi_\Lambda$ is a distribution in which parameters within a neighborhood of radius $\Lambda$, as measured by the Fisher information metric, are treated as indistinguishable and subsequently averaged together. Combining the main legs together, we realize the BRG-NNFT framework. We first construct a family of pairs $\{(\phi_\theta, \pi_\Lambda)\}_{\Lambda \in U \subseteq \mathbb{R}^+}$, where each element corresponds to a NN architecture with its parameter distribution coarse grained to a distinguishability scale $\Lambda$. We then map each pair $(\phi_\theta, \pi_\Lambda)$ into a dual SFT with action, $S_\Lambda[\phi]$. The main thesis of this paper is that the family of SFT actions $\{S_\Lambda[\phi]\}_{\Lambda \in U \subseteq \mathbb{R}^+}$ should be interpreted as an information-theoretic BRG flow.

As a proof of concept, Section 4.1 applied BRG-NNFT to infinite-width NNs, of arbitrary depth, with generic activation functions. Taking advantage of the strong constraints which are placed on such a model by the infinite-width limit and the training procedure used [102–105], this analysis is fully tractable for arbitrary NN architectures, provided one can compute the expectation values of inter-layer activation functions (66). So long as the NN is initialized as a GRP and updated via Bayes' law with Gaussian data (or equivalently trained with $L^2$ losses and standard gradient descent), the predictive distribution will remain Gaussian indefinitely throughout training [102–105]. Thus, the BRG flow of such a NN should be interpreted as

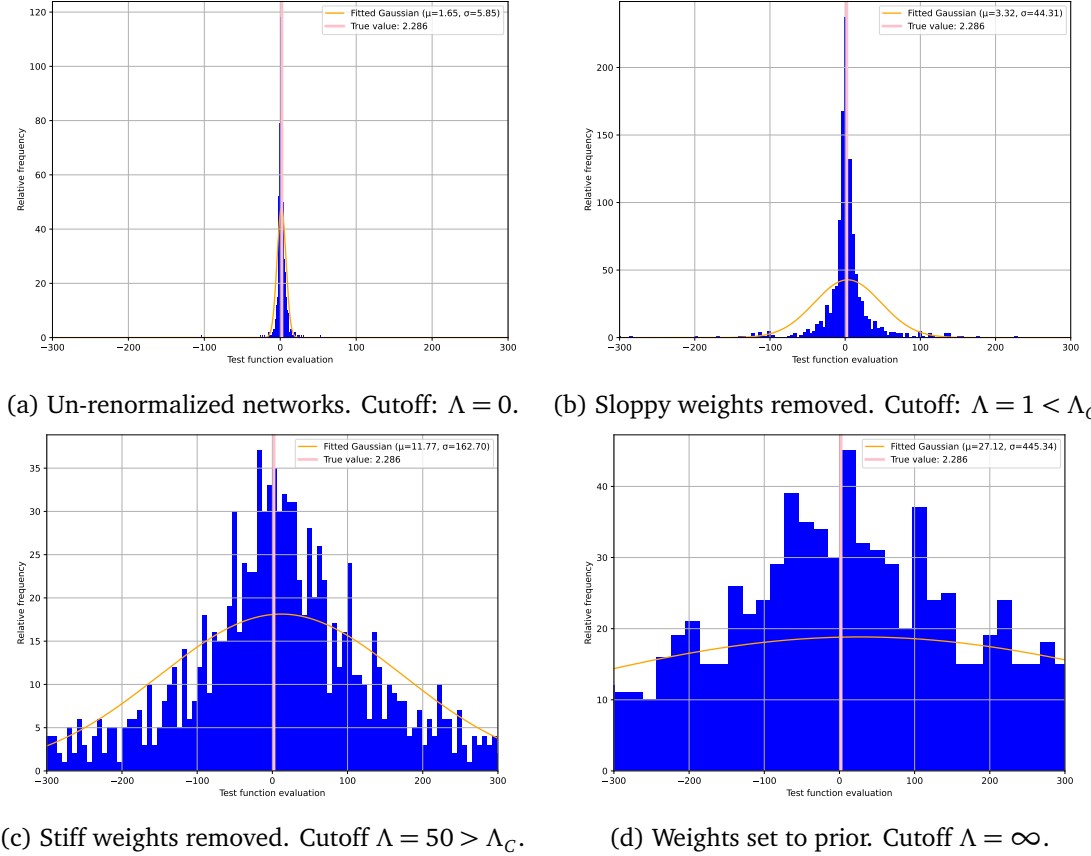

(a) Un-renormalized networks. Cutoff: $\Lambda = 0$.     (b) Sloppy weights removed. Cutoff: $\Lambda = 1 < \Lambda_C$.

(c) Stiff weights removed. Cutoff $\Lambda = 50 > \Lambda_C$.     (d) Weights set to prior. Cutoff $\Lambda = \infty$.

Figure 9: Distribution of ensemble realizations with changing cutoff. Figures 9a and 9b show the distribution of ensemble evaluations before the critical cutoff, whilst Figures 9c and 9d after $\Lambda_C$.

the analog of renormalizing a free SFT in the physics context. More precisely, the Bayesian renormalized SFT dual to the NN is a GRP with a mean and covariance that flow as a function of the distinguishability scale, $\Lambda$. The specific trajectories of these flows are determined by the Fisher information metric computed at the maximum a posteriori (MAP) parameter estimate of the fully trained model.

To better interpret the results of the general analysis undertaken in Section 4.1, Section 4.1.1 specializes to the case of a generalized cos-net architecture with scalar output. The expectation value of the generalized cos-net activation function is such that the resulting GRP is equivalent to that of a free scalar SFT with UV cutoff, $R$. Thus, in this case, the dual NNFT has a concrete physical interpretation. The BRG flow of the generalized cos-net architecture is governed by a single flowing hyperparameter, defined in (87), and equal to the trace of the covariance of the fully trained posterior distribution over parameters plus the sum of squares of its means. We refer to this hyperparameter as the effective variance of the model. At each distinguishability scale, $\Lambda$, the effective variance renormalizes the mass and momenta of the free scalar SFT (or equivalently, rescales the free field). More to the point, the BRG scheme can be regarded as renormalizing the hyperparameter $R$ which plays the role of an explicit UV cutoff in the momentum space of the free scalar SFT. In this sense, the BRG of a generalized cos-net architecture implements a genuine Wilsonian momentum-shell ERG scheme in which the UV cutoff is directly related to the information-theoretic distinguishability scale.

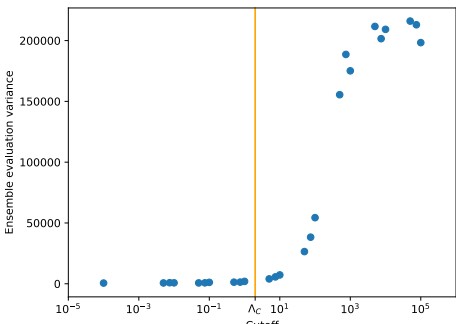
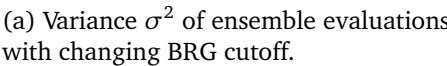
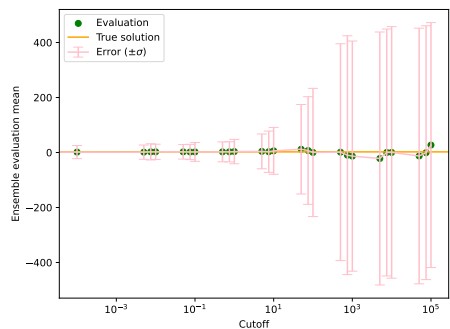

(a) Variance $\sigma^2$ of ensemble evaluations with changing BRG cutoff.

(b) Mean of ensemble evaluations with changing BRG cutoff, and error bars of $\pm\sigma$.

Figure 10: Distribution of ensemble realization characteristic statistics at fixed test data with changing cutoff.

Section 4.2 reinforces the analytic results of Section 4.1 by performing the explicit numerical training and subsequent Bayesian renormalization of an asymptotically-wide NN with ReLU activations [68]. The numerical experiment in this section illustrates several key considerations which must be taken into account if BRG-NNFT is to be implemented in practical scenarios. Principally, it is no longer feasible to express the posterior distribution over model parameters in closed form. This is the same obstacle encountered in theoretical studies of the posterior distributions of BNNs [77]. To overcome this difficulty, one can simultaneously train an ensemble of NNs with parameters randomly initialized according to a tractable distribution identifiable with the Bayesian prior [1,2,77]. The statistics of the NN ensemble after training then provide a numerical proxy for the posterior. Using NNFT one can, in principle, utilize numerically estimated $n$-point correlation functions to construct an approximate SFT representing the posterior in closed form up to a level of precision fixed by the size of the ensemble. We note that, in future work, it would be interesting to explore connections to other methods of posterior estimation in BNNs [1,2,77–80].

For the specific example at hand in Section 4.2 (i.e. an ensemble of asymptotically wide NNs trained on Gaussian data via gradient descent) we reproduce the known result [102] that the first two connected correlation functions are sufficient to construct the predictive distribution (which is a GRP) to reasonable precision both at initialization and throughout training. To Bayesian renormalize the resulting ensemble of trained NNs the Fisher information metric is computed from the terminal parameter distribution. This metric is used to guide an information-shell Bayesian renormalization scheme in which parameters are designated as sloppy or stiff at each distinguishability scale, and sloppy parameters are resampled from the prior parameter distribution. As a result, we obtain an ensemble of Bayesian renormalized NNs whose statistics are used to construct a Bayesian renormalized GRP at each scale. A distinctive feature of the BRG scheme is its identification of a critical distinguishabiliy scale up to which the performance of the model is essentially invariant under the coarse graining of sloppy parameters. At the level of the full Bayesian renormalized predictive distribution, the GRP representing the ensemble of NNs is sharply peaked for $\Lambda$ up to the critical cutoff, $\Lambda_c$, after which point the covariance increases dramatically. This result is consistent with the theoretical predictions for the flow of the mean and covariance under the BRG made in Section 4.1.

An enduring theme of this work is that of duality, both in terms of explicit analysis and in the sense of using complimentary ideas to reinforce and enrich each other. For example, on one hand, NNFT amplifies the interpretability of BRG by encoding abstract Bayesian parameter distributions in the form of more tractable SFTs. On the other hand, BRG expands the utility of NNFT by endowing it with an information-theoretic tool for navigating between SFTs. The combined framework of BRG-NNFT straddles the boundary between physics and ML. It therefore presents exciting opportunities for advancements on either side of the physics/ML collaboration as well as at their interface [109]. In this respect, BRG-NNFT is an emerging research tool that is well-situated in the landscape of physics meets ML.

In future work, we hope to push the boundaries of BRG-NNFT even further both in its capacity as a tool for improving the interpretability and theoretical tractability of AI, and in applications to exploring the space of (statistical) field theories. A particularly interesting direction centers around using BRG-NNFT as scheme for performing diffusion learning [51]. As has been described in [31], BRG implements a diffusion process via information geometrically inspired parameter coarse graining. It would be interesting to study the 'inversion' of such a diffusion process via a score based generative algorithm. Since BRG eliminates degrees of freedom in a hierarchy of scales governed by the Fisher information metric, one might suspect that reconstruction of the score vectors – the rate-limiting step in the diffusion learning paradigm – will be more efficient[31] when inverting a Bayesian diffusion process as opposed to a diffusion process which arbitrarily noises a distribution. The benefits of this alternate noising schedule are likely most apparent for complicated parameter distributions. One potential source of such distributions are those obtained from the NN duals to exotic SFTs. In this case, the parameter distribution may be analytically known but difficult to sample, meaning it would be difficult to obtain a sufficiently large ensemble of NNs. A BRG diffusion process could transform the initial intractable parameter distribution into a coarse-grained version which is easier to sample from. The score-based generative algorithm inverts the diffusion process, thereby effectively sampling the initial intractable parameter distribution. Thus, BRG-NNFT diffusion learning could provide a tool for generating samples of exotic SFTs with intractable NN parameter distributions. Furthermore, it could also be used to trace out (inverse) BRG flows for exotic SFTs.[32]

In closing, machine learning is in a period of explosive growth relative to its performance and applicability. For this reason it is crucial that we develop techniques for understanding and interpreting ML algorithms. NNFT [12, 32–34] and BRG [30, 31] are physics-inspired interpretability tools, and when brought together they provide the ingredients for mapping out the space of NNs and SFTs. In many ways, this state of affairs closely resembles the state of quantum field theory (QFT) prior to Wilson's seminal work on the renormalization group [35, 36]. At that time, QFT was a wildly powerful, but largely unwieldy tool. Wilsonian RG gave a methodology for organizing our thinking about QFTs which in turn has given us more control over how we can use and study them. Our hope is that the approach we have outlined in this paper can serve a similar organizing role for machine learning and the study of NNs.

## Code availability

The code used to obtain the numerical results in this paper can be found at the following link: https://github.com/xand-stapleton/bayes-nn-ft.

---

[31]Here, by efficient we mean both in the sense that the inversion algorithm should be less computationally expensive, and that it should retain more fine grained information.

[32]A similar approach to sampling field theories has been explored in [39].

## Acknowledgments

We thank Pietro Rotondo and Veronica Guidetti for their helpful comments on the draft. We also wish to thank Jim Halverson for useful discussions on the NNFT correspondence at String-Data 2023 and David Berman for his insight and many useful discussions over the course of this work. Finally, we thank Edward Hirst, Nathaniel Craig, Katy Craig, Emanuele Gendy, Ro Jefferson, and Zohar Ringel for useful discussions.

**Funding information** JNH was supported by the National Science Foundation under Grant No. NSF PHY-1748958 and by the Gordon and Betty Moore Foundation through Grant No. GBMF7392. MSK is supported through the Physics department at the University of Illinois at Urbana-Champaign. AM acknowledges support from Perimeter Institute, which is supported in part by the Government of Canada through the Department of Innovation, Science and Economic Development and by the Province of Ontario through the Ministry of Colleges and Universities. AGS acknowledges support from Pierre Andurand over the course of this research, and wishes to thank J. Kerrison for insightful conversations linking NNFTs and diffusion models to biological systems. This research utilised Queen Mary's Apocrita HPC facility [110], supported by QMUL Research-IT.

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
