# Peer review of "Bayesian RG Flow in Neural Network Field Theories"

_SciPost Physics Core, doi:SciPost Phys. Core 8, 027 (2025)_

## Round 2 · Referee Report · Anonymous (Referee 1) · 2024-12-30

Strengths
1. The concept is well presented. The paper is well organised and nicely written.
2. A pedagogical exposition to Bayesian renormalisation group and neural network field theory are given in the paper which is helpful for the readers, including those, who are not closely familiar with this field of research.
3. The results of the paper are summarized well. The authors have also elucidated the proposed NNFT and BRG correspondence using illustrative examples.
Weaknesses
Some Mathematical expressions and notations can be better explained to improve the clarity of the paper.
Report
The authors have explored the connection between Neural Network Field Theory correspondence and Bayesian Renormalization Group flow in the space of statistical field theories. The authors have shown that a class of SFT actions, constructed from a NN architecture with its parameter distribution coarse grained to a cut-off scale $\Lambda$, can be interpreted as an information-theoretic BRG flow. The results of the paper are significant and the paper also contains a good amount of pedagogical details about NNFT and BRG. I recommend this paper to be published in SciPost Physics Core.
Requested changes
For the convenience of the general readers it will be better if the authors can clarify some notations, which are mentioned below.
1. The target function $\phi_{\ast}$ mentioned in Table 1 in section (2.2) is not explained in the text. It will be helpful if some details of this function are discussed in the main text.
2. Could the authors comment on how $\tilde{G}^{(n)}\left(x_{1},\ldots, x_{n}\right)$ in Eq.(31) is related to $G^{(n)}\left(x_{1},\ldots, x_{n}\right)$?
3. In Eq.(32) product index should be $i$ instead of $j$.
4. Can some more details be given about the $N$-scaling in the couplings, mentioned in the paragraph below table 2? Are these couplings related to $g^{(n)}$ appearing in Eq.(35)?
5. Is the $\mathcal{N}$ appearing in Eq.(54) same as the normal density given in Eq.(21) of the footnote 11? It will be better if it is mentioned explicitly in the paragraph containing Eq.(54).
Recommendation
Publish (easily meets expectations and criteria for this Journal; among top 50%)
Author: Jessica N. Howard on 2025-02-11 [id 5212]
(in reply to Report 1 on 2024-12-30)
Thank you very much for your positive review and helpful feedback. Based on your suggestions, we have implemented the following changes:
- Thank you for catching this, we have added a sentence explaining this in Section 2.2: "Here, $\phi^*_i$ is the value of the target function $\phi^*$ which we would like our NN to approximate evaluated at $x_i$, i.e. $\phi^*_i = \phi^*(x_i)$". We have also standardized our notation to use $\phi^*$ when referring to the target function throughout the text.
- Indeed, the tilde indicates that this is a sampled (empirical) estimate of $G^{(n)}$ in Eq. 31 (which is now Eq. 37). In the limit of an infinite number of samples, $\tilde{G}^{(n)}$ should approach $G^{(n)}$. We added text clarifying this.
- Thank you for catching this typo, it has been fixed.
- Yes, exactly. We have added a short description (below what is now Eq. (43)) expanding on this connection further and discussing where in the literature more details on this topic can be found.
-
Thank you for pointing this out. Indeed, in Eq. (54) we were using a short-hand version of the same normal distribution notation seen in Eq. (21). Namely, if $y$ is a normally distributed random variable with mean $\mu$ and variance $\sigma$ the distribution can be written as $\mathcal{N}(y \mid \mu, \sigma)$. In Eq. (54) we suppressed the "$y ~\mid$" in this expression since the random variable being sampled was indicated e.g. $y \sim \mathcal{N}(\mu, \sigma)$. There is another minor difference between the two instances that we should also mention, in Eq. (21) this is assumed to be a multivariate normal distribution whereas in Eq. (54) this is a univariate normal distribution. However, since the univariate version is a special case of the multivariate version, the same $\mathcal{N}$ notation is used. Which version is meant is often gleaned from context. For example, whether one mentions a covariance $\Sigma$ (multivariate case) or a variance $\sigma$ (univariate case).
In any case, we agree that defining the notation used in Eq. (54) (now Eq. (61) ) in the text (as opposed to implicitly in a previous footnote) is an excellent suggestion. We have added text around Eq. (61) to clarify what is meant there.
We hope that the above changes have addressed your concerns. Again, we greatly appreciate your suggestions and believe they have improved the clarity of our work.
Author: Jessica N. Howard on 2025-02-11 [id 5213]
(in reply to Report 2 on 2025-01-27)Thank you very much for your positive review and helpful suggestion. Indeed, we hope that this work is amenable to a wide audience as we believe there are many potential applications. We therefore appreciate your suggestion to make some of the abstract concepts more concrete. Along that line, we have added a concrete example of Bayesian inference at the end of Section 2.1 which we hope better illustrates some of the abstract concepts discussed.

---

## Round 2 · Referee Report · Anonymous (Referee 2) · 2025-1-27

Strengths
1. The paper is very well-written, clear and the logic is easy to follow.
2. The paper pedagogically introduces the necessary preliminaries like Bayesian inference, Bayesian RG, the NNFT correspondence.
3. The paper demonstrates the proposed BRG-NNFT formalism through clearly, explained analytical and numerical examples.
Weaknesses
1. The mathematical notation in sections where Bayesian Inference and Bayesian RG is discussed is a bit difficult to follow for a non-expert.
Report
The paper applies the formalism of Bayesian Renormalization Group (BRG) to perform an information theoretic coarse graining in the parameter space of neural networks. This leads to a flow in the parameter space which via the NN-FT formalism, is interpreted as a flow in the space of SFTs from the information-theoretic UV (fully trained network) to IR (untrained network). The training of a neural network in this language is an inverse BRG flow. The proposed BRG-NNFT correspondence is then substantiated through concrete analytical and numerical examples. The paper is an important contribution to the topic of NN-FT correspondence and can shed light on an understanding of how neural networks learn. I strongly recommend it to be published in SciPost Physics Core.
Requested changes
1. No important change requested. Some more pedagogical explanation of the concepts of Bayesian inference and Bayesian RG will be useful for a non-reader.
Recommendation
Publish (easily meets expectations and criteria for this Journal; among top 50%)

---

## Round 3 · Author Response

We greatly appreciate the feedback of the referees and have implemented changes to address their suggestions. We believe that this has improved the clarity of this work and has hopefully helped make it amenable to a wider audience. We provide a list of changes below.

---

## Round 3 · List of Changes

Below is a point-by-point list of changes in response to Referee 1’s feedback: 1. We have added a sentence explaining the meaning of $\phi^*$ in Section 2.2: "Here, $\phi^*_i$ is the value of the target function $\phi^*$ which we would like our NN to approximate evaluated at $x_i$, i.e. $\phi^_i = \phi^(x_i)$". We have also standardized our notation to use $\phi^*$ when referring to the target function throughout the text. 2. We added text clarifying that the tilde on $\tilde{G}^{(n)}$ indicates that this is a sampled (empirical) estimate of $G^{(n)}$ in Eq. 31 (which is now Eq. 37). In the limit of an infinite number of samples, $\tilde{G}^{(n)}$ should approach $G^{(n)}$. 3. We appreciate Referee 1 catching the typo in Eq. 32 (now Eq. 39), it has been fixed. 4. We have added a short description (below what is now Eq. (43)) expanding on the connection between the connected correlation functions and the couplings $g^{(n)}$ and discussing where in the literature more details on this topic can be found. 5. As suggested by Referee 1, we have better defined the notation used for normally distributed random variables around what is now Eq. 61.

In response to Referee 2’s feedback, we also added a concrete example of Bayesian inference at the end of Section 2.1 which we hope illustrates some of the abstract concepts discussed.

We hope that these changes adequately addressed the referees’ concerns and believe that they have helped improve the clarity of our work.

---

## Editorial Decision

published